# Membrane potential states gate synaptic consolidation in human neocortical tissue

Franz X. Mittermaier [1], Thilo Kalbhenn [2], Ran Xu[3], Julia Onken[3], Katharina Faust[3], Thomas Sauvigny [4], Ulrich W. Thomale[5], Angela M. Kaindl[6], Martin Holtkamp [7], Sabine Grosser [8], Pawel Fidzinski [9], Matthias Simon [2], Henrik Alle [1] & Jörg R. P. Geiger[1] ✉

Synaptic mechanisms that contribute to human memory consolidation remain largely unexplored. Consolidation critically relies on sleep. During slow wave sleep, neurons exhibit characteristic membrane potential oscillations known as UP and DOWN states. Coupling of memory reactivation to these slow oscillations promotes consolidation, though the underlying mechanisms remain elusive. Here, we performed axonal and multineuron patch-clamp recordings in acute human brain slices, obtained from neurosurgeries, to show that sleep-like UP and DOWN states modulate axonal action potentials and temporarily enhance synaptic transmission between neocortical pyramidal neurons. Synaptic enhancement by UP and DOWN state sequences facilitates recruitment of postsynaptic action potentials, which in turn results in long-term stabilization of synaptic strength. In contrast, synapses undergo lasting depression if presynaptic neurons fail to recruit postsynaptic action potentials. Our study offers a mechanistic explanation for how coupling of neural activity to slow waves can cause synaptic consolidation, with potential implications for brain stimulation strategies targeting memory performance.

As humans, we have the ability to recall detailed information, even from years in the past, indicating a powerful memory system. Newly encoded explicit memories initially depend on the hippocampus[1–4]. Memory reactivation, mediated by a hippocampo-cortical dialog, leads to a gradual maturation of neocortical engrams over time[5–9]. After this systems consolidation process, the neocortex can store information for decades.

It is well established that consolidation relies on non-rapid eye movement (NREM) sleep[10–15]. This brain state gives rise to characteristic patterns in the electroencephalogram, including slow waves (~0.5–4 Hz), sleep spindles (~10–16 Hz) and hippocampal ripple oscillations (~80–120 Hz in humans)[16–18]. During slow wave activity (SWA), neocortical neurons exhibit synchronous membrane potential changes, referred to as UP and DOWN states[19–22]. UP states are periods

[1]Charité—Universitätsmedizin Berlin, corporate member of Freie Universität Berlin and Humboldt-Universität zu Berlin, Institute of Neurophysiology, Berlin, Germany. [2]Department of Neurosurgery (Evangelisches Klinikum Bethel), University of Bielefeld Medical Center OWL, Bielefeld, Germany. [3]Department of Neurosurgery, Charité—Universitätsmedizin Berlin, corporate member of Freie Universität Berlin and Humboldt-Universität zu Berlin, Berlin, Germany. [4]Department of Neurosurgery, University Medical Center Hamburg-Eppendorf, Hamburg, Germany. [5]Pediatric Neurosurgery, Charité—Universitätsmedizin Berlin, corporate member of Freie Universität Berlin and Humboldt-Universität zu Berlin, Berlin, Germany. [6]Department of Pediatric Neurology, Charité—Universitätsmedizin Berlin, corporate member of Freie Universität Berlin and Humboldt-Universität zu Berlin, Berlin, Germany. [7]Department of Neurology, Charité—Universitätsmedizin Berlin, corporate member of Freie Universität Berlin and Humboldt-Universität zu Berlin, Berlin, Germany. [8]Institute for Integrative Neuroanatomy, Charité—Universitätsmedizin Berlin, corporate member of Freie Universität Berlin and Humboldt-Universität zu Berlin, Berlin, Germany. [9]Neuroscience Clinical Research Center, Charité—Universitätsmedizin Berlin, corporate member of Freie Universität Berlin and Humboldt-Universität zu Berlin, NeuroCure Cluster of Excellence, Berlin, Germany. ✉e-mail: joerg.geiger@charite.de

of increased neural activity, giving rise to depolarization of neurons[23,24]. Conversely, DOWN states are silent periods, associated with hyperpolarization[25,26]. In the human neocortex, prominent SWA occurs in supragranular layers 2 & 3[21,27]. Several studies have demonstrated that precise temporal coupling of spindles and ripples to SWA promotes engram reactivation[28–34] and determines success of memory consolidation[18,35–38]. Consequently, brain stimulation methods that boost SWA or enhance coupling have a positive effect on memory performance in rodents and humans[39–44]. These observations suggest that SWA and the underlying membrane potential UP and DOWN states initiate mechanisms that augment memory functions. However, in the human brain such mechanisms remain elusive.

One possibility is that UP and DOWN states modulate excitatory synapses in the neocortex to increase synaptic strength during SWA-coupled neural activity. While action potentials (AP) are necessary to initiate transmission in the mammalian neocortex, it has been demonstrated in laboratory animals that presynaptic signals below the AP-threshold (i.e., subthreshold signals) have a modulatory effect on synaptic strength[45–54]. For instance, at synapses between neocortical pyramidal neurons in ferrets[46] and rats[47] a > 1-second-long subthreshold depolarization preceding an AP leads to an increase in synaptic amplitude. Through such mechanisms, UP and DOWN states could tune local synaptic networks to promote long-term synaptic plasticity, which is believed to be fundamental for memory consolidation[2,55].

Therefore, we set out (1) to investigate whether subthreshold signals, mimicking UP and DOWN states, modulate synaptic transmission in human supragranular neocortical layers, and (2) to explore the mechanisms through which this modulation could promote synaptic consolidation.

To address this, acute brain slices were prepared from neocortical samples that were resected during neurosurgery[56]. Our multineuron patch-clamp approach[57] was used to (1) record from pairs of synaptically connected neurons, (2) both the soma and axon of individual neurons and (3) multiple neurons within the intact human layer 2 & 3 microcircuit. We found that presynaptic membrane potential depolarizations, mimicking UP states during SWA, lead to inactivation of axonal voltage-gated potassium channels ($K_v$), which causes broadening of axonal APs. Broadened axonal APs, in turn, increase presynaptic reliability at proximal synapses between human layer 2 & 3 pyramidal neurons. Sequences of de- and hyperpolarizations, which resemble UP → DOWN → UP cycles, can additionally recover voltage-gated sodium channels ($Na_v$) from inactivation, which increases the amplitude of axonal APs and further facilitates synaptic transmission. Due to this increase in synaptic strength, neural activity recruits postsynaptic APs more reliably when it is coupled to DOWN-to-UP transitions during UP → DOWN → UP cycles. Finally, we found that such postsynaptic APs are a prerequisite for lasting stabilization of synaptic strength. Taken together, these results provide a mechanistic

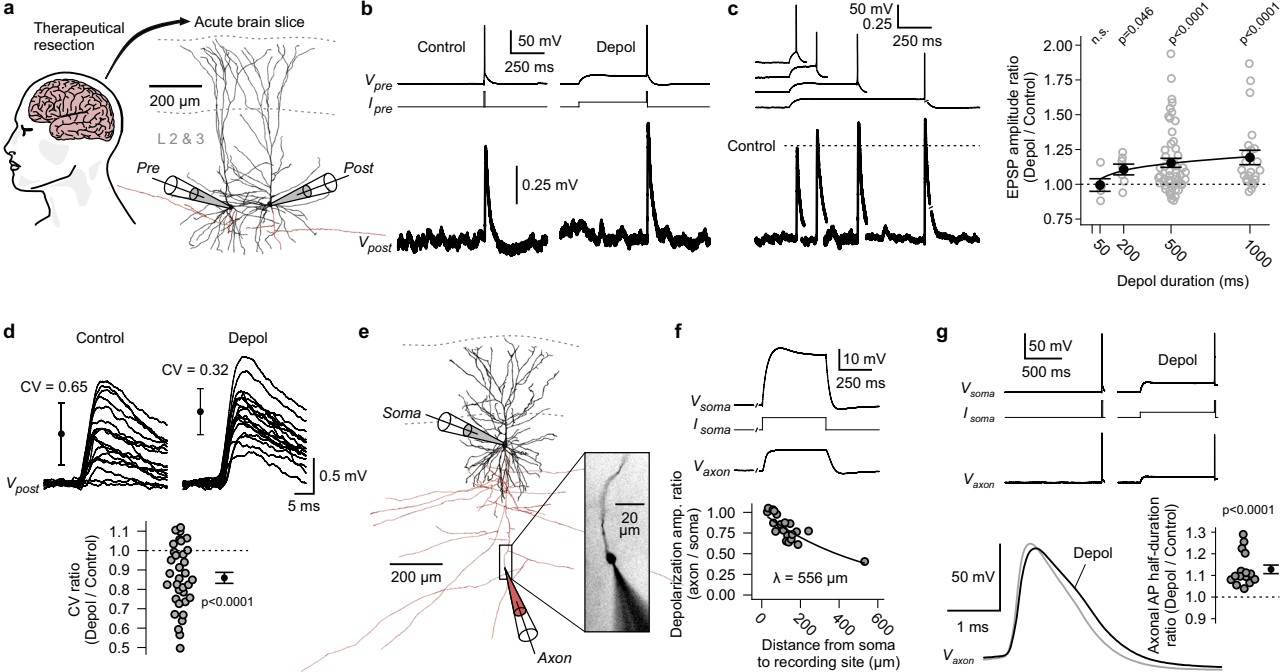

**Fig. 1 | Presynaptic subthreshold depolarizations increase synaptic strength through broadening of axonal action potentials. a** Schematic of the human neocortex and reconstruction of two pyramidal neurons, which were connected by a unitary synapse (*Pre* and *Post* stands for pre- and postsynaptic neuron). **b** Paired whole-cell patch-clamp recording of neurons shown in panel a. Current was injected into the presynaptic neuron to elicit action potentials (AP) from resting membrane potential ('Control') or after a subthreshold depolarization ('Depol'). Excitatory postsynaptic potentials (EPSP) were averaged over multiple trials. **c** Four separate recordings with different durations of depolarizations. To visualize relative change of EPSP amplitudes, postsynaptic signals were normalized to the 'Control' condition of each recording (dotted line). Right, relative changes of EPSP amplitudes plotted against depolarization durations (p-values were computed using two-sided Wilcoxon signed-rank tests; 50 ms, n = 5 paired recordings; 200 ms, n = 7; 500 ms, n = 53; 1000 ms, n = 23; error bars show mean ± s.e.m; fit line corresponds to bi-exponential function with time-constants for $K_v$1-inactivation, see Fig. 2d). **d** Top, single-trial EPSPs of an exemplary synapse (error bars show

mean ± s.d. of single-trial amplitudes; traces were smoothed using a moving average with a 1 ms window; CV: coefficient of variation). Bottom, summary plot of relative changes of CVs (experiments with 500 or 1000 ms 'Depol'-duration and effect > 1.1 in panel c were pooled; n = 34 paired recordings; two-sided Wilcoxon signed-rank test; error bar shows mean ± s.e.m.). **e** Reconstruction of exemplary somato-axonal recording. Inset, axonal 'bleb' filled with Alexa Fluor 568 dye during recording. **f** Somato-axonal recording of the neuron shown in panel e. Current was injected at the soma to cause a subthreshold depolarization, which spread into the axon. Bottom, relationship between distances from soma and attenuation of amplitudes of passively spreading depolarizations (n = 20 somato-axonal recordings; mono-exponential fit). **g** Somato-axonal recording. APs were elicited at the soma. Bottom left, overlay of 'Control' and 'Depol' AP recorded in the axon. Bottom right, summary plot of relative changes of axonal AP half-durations (n = 15 somato-axonal recordings; two-sided Wilcoxon signed-rank test; error bar shows mean ± s.e.m.). Source data are provided as a Source Data file.

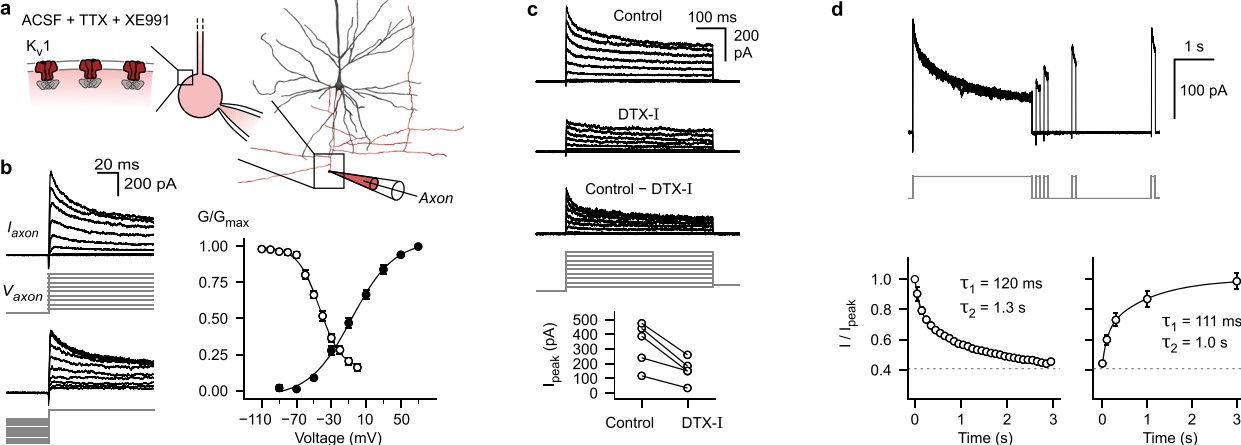

**Fig. 2 | Axons of human layer 2 & 3 pyramidal neurons contain $K_v 1$ potassium channels that show slow inactivation in the subthreshold voltage range.** **a** Reconstruction of biocytin labeled pyramidal neuron (dendrite black, axon red) and schematic of 'whole-bleb' recording configuration (see Methods: Axonal voltage-clamp recordings). **b** Left, $K^+$-currents in response to activation and steady-state inactivation voltage-clamp protocols (see Methods). Right, normalized $K^+$-conductance as a function of test voltage (black data points, activation protocol, n = 9 axonal recordings) or conditioning voltage (white data points, inactivation protocol, n = 5 axonal recordings). Data is displayed as mean ± s.e.m. **c** $K^+$-currents

in response to voltage steps during control period, after local puff-application of dendrotoxin-I (DTX-I), as well as subtraction of the two conditions. Bottom, summary plot showing peak currents in response to +70 mV test pulses before and after DTX-I application (n = 5 axonal recordings). **d** $K^+$-currents in response to a protocol to determine inactivation and recovery from inactivation kinetics (see Methods). Bottom, summary plots displaying mean ± s.e.m of normalized current amplitudes (n = 5 axonal recordings). Lines correspond to double exponential fits (time constants are shown as insets). ACSF, artificial cerebrospinal fluid; TTX, tetrodotoxin. Source data are provided as a Source Data file.

framework that can explain the consistent in-vivo observation that coupling of neural activity to SWA leads to an enhancement of memory consolidation capabilities in humans[15].

## Results

### Modulation of synaptic transmission by presynaptic subthreshold depolarizations

To address whether subthreshold membrane potential changes modulate synaptic transmission in the human neocortex, we studied synapses between pyramidal neurons in layers 2 & 3 of neocortical brain slices (Fig. 1a, see Methods). Tissue samples were obtained from patients who underwent neurosurgery for the treatment of drug-resistant epilepsy (n = 33) or brain tumors (n = 12, see Methods). We performed paired somatic whole-cell patch-clamp recordings of synaptically connected pyramidal neurons. APs were elicited in presynaptic neurons either from resting membrane potential (−76 ± 5 mV, mean ± s.d., 'Control' condition) or following a subthreshold depolarization (−56 ± 5 mV, mean ± s.d., 'Depol' condition, Fig. 1b). Amplitudes of excitatory postsynaptic potentials (EPSPs) were increased if APs were preceded by a presynaptic subthreshold depolarization (Fig. 1b). The extent of this synaptic enhancement depended on the duration of the depolarization (Fig. 1c). While depolarizations with a length of 50 ms did not cause an increase in average EPSP amplitude, those with a length of 200, 500 and 1000 ms led to a mean increase of +11 ± 4%, +15 ± 3% and +19 ± 5% (mean ± s.e.m.), respectively (Fig. 1c). In addition to sufficient duration, depolarization amplitudes had to be >10 mV (measured at the soma) to reliably cause enhancement of synaptic transmission (Supplementary Fig. 1). Five seconds after the end of a subthreshold depolarization, EPSP amplitudes had returned to baseline values, indicating a short-term enhancement (Supplementary Fig. 2). Notably, we observed this synaptic effect in tissue samples from patients with refractory epilepsy as well as patients with brain tumors, the latter group consisting of patients with- or without documented seizures (Supplementary Fig. 3). While most experiments were performed in tissue samples from the temporal neocortex, the effect was also found in frontal and parietal tissue samples (Supplementary Fig. 3). Analysis of single-trial EPSPs demonstrated that the

enhancement of synaptic transmission can be attributed to an increase in synaptic reliability. Specifically, the coefficient of variation (CV) of single-trial EPSP amplitudes was significantly reduced in the 'Depol' condition (n = 34 paired recordings; two-sided Wilcoxon signed-rank test, p < 0.0001; Fig. 1d), indicative of a presynaptic mechanism.

How do presynaptic subthreshold depolarizations increase the reliability of synaptic release? Studies in rats and ferrets have found that subthreshold signals, originating at the dendrites or soma, passively spread along the axon for a certain distance[45,46]. Through inactivation of $K_v$ channels, such depolarizations can lead to the broadening of axonal APs[46–49,58]. The broadening of axonal APs, in turn, results in the increase of presynaptic calcium influx and consequently leads to enhancement of the synaptic release probability[49,59,60]. To test whether subthreshold depolarizations also lead to broadening of APs in human axons, we conducted paired somato-axonal patch-clamp recordings in layer 2 & 3 pyramidal neurons ('bleb'-recordings[46], see Methods, Fig. 1e). Amplitudes of subthreshold depolarizations, induced by somatic current injection, attenuated along the axons with a length constant of ~550 µm, due to the cable properties of these fine structures (Fig. 1f). This length constant suggests that such depolarizations can reach presynaptic terminals of local synapses[61]. APs were elicited at the soma and propagated to the recording sites in the axons (Fig. 1g). Axonal APs, which were preceded by a 1000 ms subthreshold depolarization, had a significantly increased half-duration (+13 ± 2%, mean ± s.e.m.; n = 15 somato-axonal recordings; two-sided Wilcoxon signed-rank test, p < 0.0001; Fig. 1g).

To establish whether inactivation of $K_v$-channels[46–49] is the likely mechanism for broadening of axonal APs in human pyramidal neurons, voltage-clamp recordings in the 'whole-bleb' configuration were conducted (Fig. 2a). These experiments revealed an outward potassium current that activated at negative voltages (half-maximal activation at −7 mV, slope factor of 24, Fig. 2b; membrane voltages were not corrected for liquid junction potential of −15 mV, see Methods). The channels mediating this current displayed inactivation (half-maximal inactivation at −43 mV, slope factor of 13, Fig. 2b). While they were almost completely available for activation at the typical axonal resting membrane potential (~ − 75 mV), they underwent considerable steady-

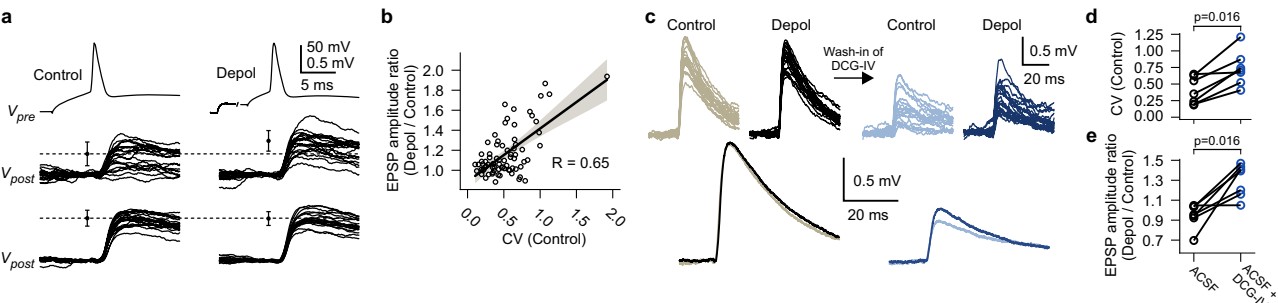

**Fig. 3 | Synaptic reliability determines magnitude of subthreshold modulation.**
**a** Multiple overlaid single-trial excitatory postsynaptic potentials (EPSP) of one unreliable synapse (i.e., large CV of single-trial EPSP amplitudes during 'Control' condition, top) and one reliable synapse (i.e., small CV, bottom). Note enhancement of unreliable synapse by 'Depol' condition (presynaptic traces are only shown for one of the two synapses; error bars show mean ± s.d. of n = 20 single-trial EPSP amplitudes; traces were smoothed using a moving average with a 1 ms window).
**b** Relative changes of EPSP amplitudes in response to 'Depol' conditions plotted against CVs of single trial EPSP amplitudes during 'Control' conditions

(experiments with 500 and 1000 ms 'Depol'-duration were pooled; n = 76 paired recordings; line and error band represent regression line and 95% confidence interval of a linear model). **c** Top, single-trial EPSPs of an exemplary synapse before and after wash-in of 100 nM DCG-IV. Bottom, averaged EPSPs. **d, e** Summary plots depicting decrease of synaptic reliability and increase of enhancement by subthreshold depolarizations in presence of DCG-IV (n = 7 paired recordings; two-sided Wilcoxon signed-rank tests; ACSF, artificial cerebrospinal fluid). Source data are provided as a Source Data file.

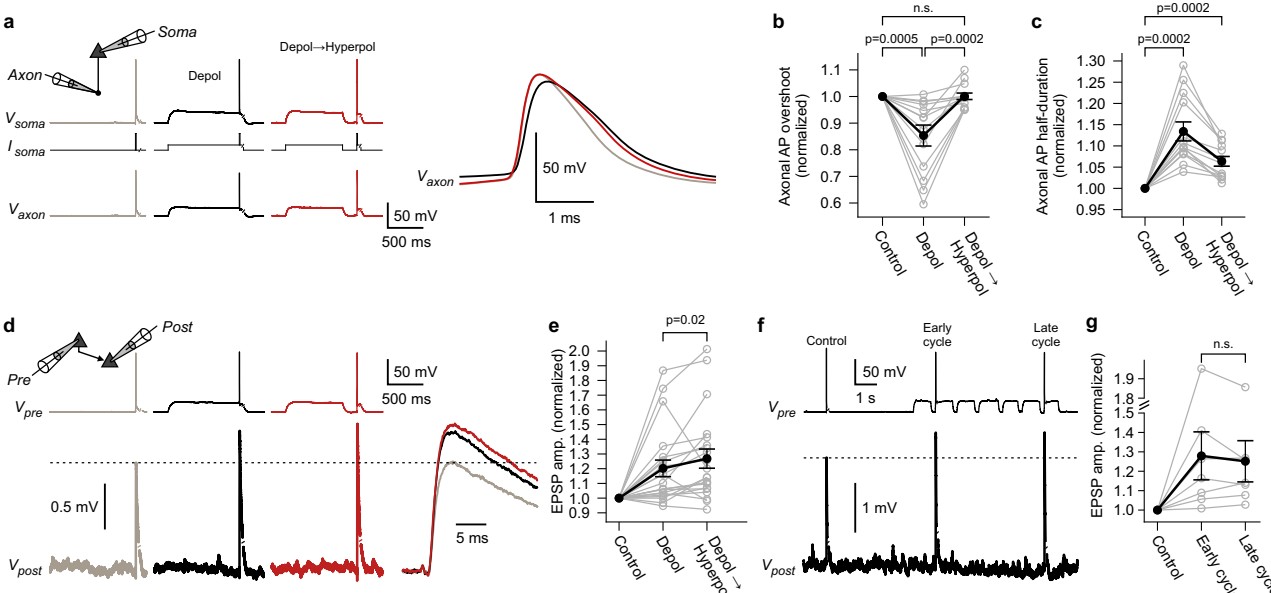

**Fig. 4 | Sequences of presynaptic de- and hyperpolarizations recover axonal action potential amplitude and further increase synaptic transmission.**
**a** Exemplary somato-axonal recording. Somatic current injections ($I_{soma}$) were used to induce 'Depol' and 'Depol→Hyperpol' conditions. Right, magnified axonal action potentials (AP). Note the decrease and rescue of axonal AP overshoot in the 'Depol' and 'Depol→Hyperpol' conditions, respectively. **b, c** Summary graphs showing relative changes of the axonal AP overshoot and half-duration (n = 13 somato-axonal recordings; two-sided Wilcoxon signed-rank tests; error bars show mean ± s.e.m). **d** Exemplary paired recording of synaptically connected pyramidal neurons.

Excitatory postsynaptic potentials (EPSP) were averaged over multiple trials and are shown on a finer timescale on the right. **e** Summary graph showing relative changes of EPSP amplitudes (n = 21 paired recordings; two-sided Wilcoxon signed-rank test; error bars show mean ± s.e.m). **f** Exemplary paired recording. 'Control' condition was compared to early and late cycles of multiple de- and hyperpolarizations.
**g** Summary graph showing relative changes of the average EPSP amplitudes (n = 7 paired recordings; two-sided Wilcoxon signed-rank test; error bars show mean ± s.e.m). Source data are provided as a Source Data file.

state inactivation in the voltage range to the AP threshold (∼ − 45 mV, Fig. 2b). This suggests that the availability of these channels can be altered by subthreshold signals. Local puff-application of dendrotoxin-I (DTX-I) reduced peak currents by 55% ± 6% (mean ± s.e.m., Fig. 2c). The DTX-I sensitive fraction of the current corresponded to the inactivating component. This indicates that the observed inactivating current is largely mediated by potassium channels containing $K_v1$ alpha subunits (Fig. 2c)[62]. The time course of inactivation in response to 3 s long steps to +20 mV was slow and best approximated by a sum of two exponential functions ($\tau_1 = 120$ ms, $\tau_2 = 1.3$ s, Fig. 2d)[63]. The recovery from inactivation after such prolonged steps also followed a

slow time course ($\tau_1 = 111$ ms, $\tau_2 = 1.0$ s, Fig. 2d). Taken together, subthreshold depolarizations can efficiently spread along the axon of human layer 2 & 3 pyramidal neurons, lead to broadening of axonal APs, through inactivation of $K_v1$ channels, and enhance synaptic transmission between local pyramidal neurons.

While a substantial fraction of synapses displayed enhancement in response to presynaptic subthreshold depolarizations, others were unaffected (Fig. 1c). This variability could largely be explained by a strong correlation between the reliability of a synapse during the 'Control' condition and the magnitude of enhancement (Pearsons R = 0.65; n = 76 paired recordings; Correlation test based on t-statistic,

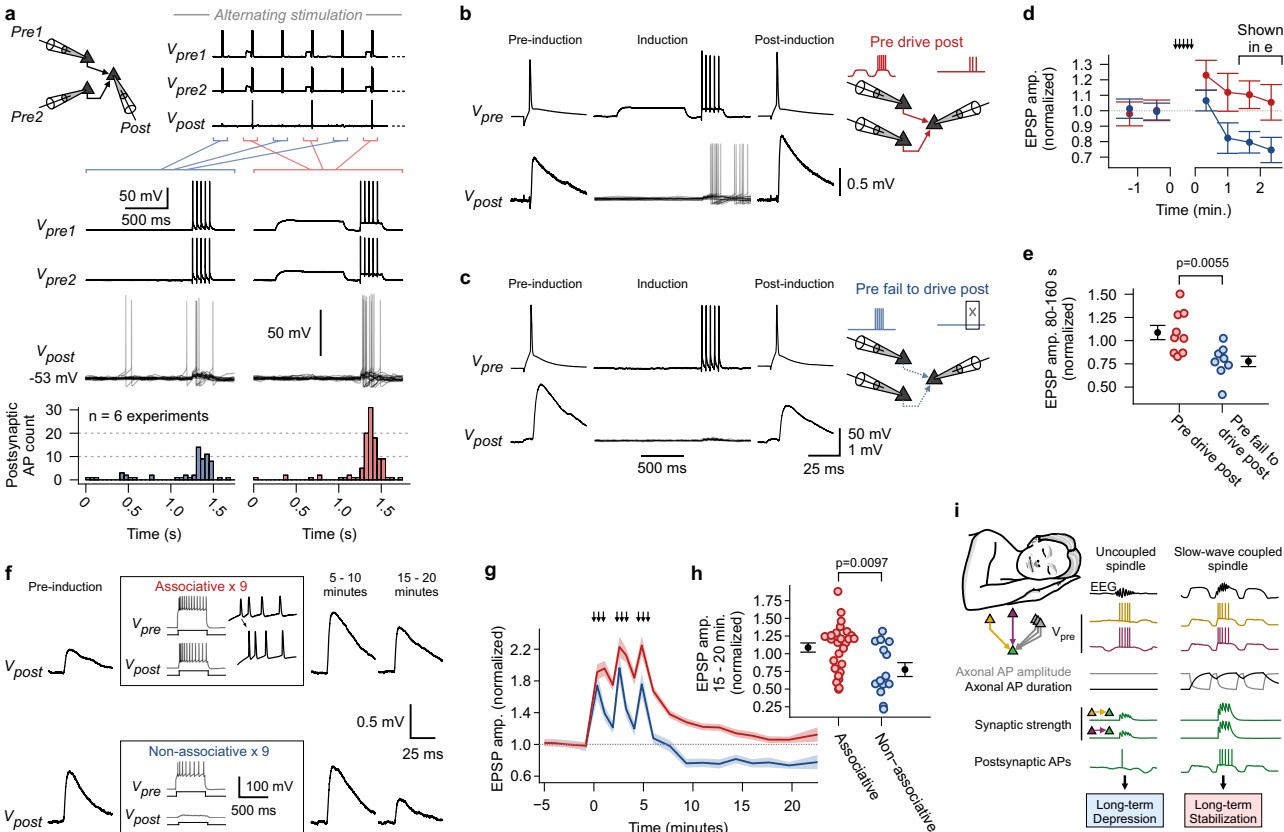

**Fig. 5 | Sequences of presynaptic de- and hyperpolarizations boost recruitment of postsynaptic action potentials, resulting in lasting stabilization of synapses.** **a** Top: Schematic depicting multineuron patch-clamp recording of convergent synaptic motif. 'Control' and 'Depol→Hyperpol' conditions followed by trains of action potentials (AP) were induced in an alternating fashion in presynaptic neurons. Bottom: First two rows show exemplary voltage traces of presynaptic neurons. Third row depicts superimposed trials of postsynaptic traces. Fourth row shows summary histogram (n = 6 experiments; 50 ms bins). **b** Presynaptic AP and averaged excitatory postsynaptic potential (EPSP) of an exemplary synapse before and after 'Pre drive post' paradigm, which consisted of synchronous initiation of the 'Depol→Hyperpol' protocol in presynaptic neurons of a convergent motif to trigger postsynaptic APs, as shown in superimposed traces (sustained current was injected into postsynaptic neuron during induction; induction was typically repeated 25 times). **c** 'Pre fail to drive post' paradigm, where postsynaptic neurons were held at a more negative membrane potential, so that synapses didn't trigger postsynaptic APs. **d** Mean EPSP

amplitudes plotted over time for the two paradigms (normalized to pre-induction period; mean ± s.e.m.). **e** EPSP amplitudes for time window 80–160 s after end of induction (normalized to pre-induction period; n = 9 synapses for 'Pre drive post', n = 8 synapses for 'Pre fail to drive post'; two-sided Mann-Whitney U test; mean ± s.e.m.). **f** Averaged EPSPs of two exemplary synapses before induction, as well as 5–10 and 15–20 minutes after start of induction. Paradigms consisted of 500-ms current injections to elicit associated pre- and postsynaptic trains of APs ('Associative') or isolated presynaptic trains of APs ('Non-associative'). The induction was repeated 9 times over a period of 5 minutes (arrows in panel g). **g** Mean EPSP amplitudes plotted over time for the two paradigms (normalized to pre-induction period; mean ± s.e.m.). **h** EPSP amplitudes for time window 15–20 minutes (normalized to pre-induction period; n = 27 synapses for 'Associative', n = 14 synapses for 'Non-associative'; two-sided Mann-Whitney U test; mean ± s.e.m.). **i** Concept of coupled vs. uncoupled sleep oscillations and their short- and long-term effect on synaptic strength. Source data are provided as a Source Data file.

p < 0.0001; Fig. 3a, b). If a synapse had a higher CV of single-trial EPSP amplitudes during 'Control' condition (i.e., the synapse was less reliable), then it was facilitated stronger during the 'Depol' condition (Fig. 3a, b). If a synapse was already highly reliable, then it was not further enhanced (Fig. 3a, b). We used DCG-IV, a group II metabotropic glutamate receptor agonist, to reduce the release probability of highly reliable synapses (Fig. 3c). Remarkably, in the presence of DCG-IV, such synapses switched to display enhancement by presynaptic subthreshold depolarizations (Fig. 3c–e). This indicates that subthreshold modulation likely occurs at most synapses between pyramidal neurons in neocortical layers 2 & 3, but the extent of enhancement depends on the release probability, which amongst others can be set by neuromodulation, extracellular calcium concentration[64] or extracellular glutamate concentration[65].

## Sequences of presynaptic de- and hyperpolarizations

Depolarizations ('Depol' condition) caused broadening of axonal APs, but also reduced axonal AP amplitudes (−15 ± 4% for AP overshoot, mean ± s.e.m.; n = 13 somato-axonal recordings; two-sided Wilcoxon

signed-rank test, p = 0.0005; Fig. 4a, b). Based on results from rodent studies[66,67] and recent evidence on properties of ion channels in human neocortical pyramidal neurons[68], it is plausible that this reduction is due to inactivation of Na$_v$ channels. To promote recovery of Na$_v$s, we modified our stimulation paradigm. A sequence of a subthreshold depolarization (800 ms), followed by a brief episode (200 ms) during which a neuron repolarized to its resting membrane potential, was induced ('Depol→Hyperpol' condition, Fig. 4a). As expected, paired somato-axonal recordings revealed that this paradigm restores the amplitude of axonal APs (Fig. 4b). While the amplitude was completely restored, the half-duration of axonal APs remained broadened compared to 'Control' condition (Fig. 4c). This result is in line with the slow recovery from inactivation of axonal K$_v$1 potassium channels (Fig. 2d). In summary: When subthreshold depolarizations are followed by short hyperpolarizations, fast recovery from inactivation of Na$_v$s (time constant of ~10 ms[68]) outpaces slow recovery of axonal K$_v$1 channels, leading to tuned axonal APs (Supplementary Fig. 4). This raised the question of whether such tuned axonal APs lead to further enhancement of synaptic transmission, as has been found in rodents[66]. To

investigate this, paired somatic recordings of synaptically connected pyramidal neurons were conducted (Fig. 4d). Presynaptic sequences of a de- and hyperpolarizations typically induced a further enhancement of synaptic strength compared to simple depolarizations (further enhancement by a mean of +8% to a total enhancement of +27%; n = 21 paired recordings; two-sided Wilcoxon signed-rank test, p = 0.02; Fig. 4d, e). Multiple UP-DOWN cycles had the same effect on synaptic transmission as a single sequence of a de- and brief hyperpolarization (Fig. 4f, g). We also studied the effect of isolated hyperpolarizations (without preceding depolarizations, Supplementary Fig. 5). Isolated 200 ms long hyperpolarizations increased AP amplitudes only in those axons that had resting membrane potentials more depolarized than −77 mV (5/10 axons, Supplementary Fig. 5a–c), suggesting that these axons contain inactivated $Na_v$s at resting. Similar to what has been found in rodents[66], isolated hyperpolarizations enhanced synaptic strength by +12 ± 4% (mean ± s.e.m.; n = 23 paired recordings; two-sided Wilcoxon signed-rank test, p = 0.012; Supplementary Fig. 5d, e). Taken together, both isolated de- and hyperpolarizations can separately modulate axonal AP shape and synaptic transmission, but sequences of de- and brief hyperpolarizations represent the ideal stimulus to boost synaptic strength. In the context of sleep, such sequences are of particular interest, as neocortical neurons cycle through membrane potential UP and DOWN states during SWA[19–22]. Our results suggest that UP-state-like depolarizations can enhance transmission, but sequences of UP → DOWN → UP states further increase synaptic strength. This implies that periods of SWA represent fine-tuned time windows of augmented presynaptic potency.

## Boosting synaptic plasticity induction

UP and DOWN states in the intact brain occur synchronously in many neighboring neurons. To test the potential impact of presynaptic APs during synchronized UP and DOWN states across several neurons, we conducted multineuron patch-clamp recordings[57]. Convergent synaptic motifs, i.e., two to three presynaptic pyramidal neurons that form synapses onto a single postsynaptic pyramidal neuron, were identified (Fig. 5a). Synchronized de- and hyperpolarizations were induced in the presynaptic neurons followed by a train of 5 APs. Additionally, sustained current injection was applied to keep the membrane potential of the postsynaptic neuron close to its AP-threshold. In comparison to trains of presynaptic APs, which were elicited from the resting membrane potential, those with preceding de- and hyperpolarizations triggered more postsynaptic APs (across all n = 6 experiments, postsynaptic APs were triggered in 44/163 trials for 'Control' condition and in 83/163 trials for 'Depol→Hyperpol' condition; Fisher exact test, p < 0.0001; Fig. 5a). This implies that synchronous UP → DOWN → UP sequences, along with a precise timing of presynaptic APs, enhance the recruitment of APs in postsynaptic neurons in local neocortical networks[27].

In a separate set of experiments, EPSP amplitudes of the synapses, which made up convergent motifs, were recorded as a first step (Fig. 5b). Subsequently, synchronous presynaptic de- and hyperpolarizations were induced to drive APs in the postsynaptic neuron ('Pre drive post' paradigm, Fig. 5b). After this 'Pre drive post' paradigm, the synapses displayed a potentiation to a mean of 123 ± 10% (mean ± s.e.m.), which differed from the short-term effect caused by subthreshold signals (Figs. 1–4), as it decayed slowly during the recording time of ~2.5 minutes (Fig. 5d). Conversely, a clear and lasting reduction in synaptic strength to 75 ± 8% (mean ± s.e.m.) was observed when presynaptic neurons were unable to induce firing in the postsynaptic neuron ('Pre fail to drive post', Fig. 5c, d). This suggests that associative and non-associative plasticity occurs at human synapses between layer 2 & 3 pyramidal neurons[69,70] (Fig. 5e).

To substantiate these observations, we systematically investigated synaptic plasticity utilizing paired patch-clamp recordings of connected pyramidal neurons[71], under a physiological extracellular

calcium concentration of 1.2 mM[72] (Fig. 5f). Following a 5-minute baseline, 500-ms suprathreshold step currents were injected into the pre- and postsynaptic neurons to elicit associated trains of APs ('Associative' paradigm, Fig. 5f). The frequency of the induced AP trains was ~30 Hz (32.5 ± 17 Hz, mean ± s.d.). The presynaptic neuron typically led the postsynaptic neuron for the first AP (no attempt was made to control the order of the following APs). Over the course of a 5-minute induction period, these associated trains of APs were repeated nine times (Fig. 5f, g). Following the induction, an initial post-tetanic potentiation of the synapses to a mean of 187% of the baseline amplitude occurred (Fig. 5g). The potentiation then approached a mean value of 108 ± 6% (mean ± s.e.m.), 15 minutes after the induction had ended (Fig. 5g). This mean increase to 108% (mean across all synapses for the 15–20-minute bin) failed to reach statistical significance (n = 27 paired recordings; two-sided Wilcoxon signed-rank test, p = 0.3). However, synapses displayed a large variability in their response to the plasticity induction protocol, with some showing a clear and lasting potentiation (Supplementary Fig. 6). The overall stabilization of synaptic strength depended on NMDA-receptors and was converted to lasting depression when D-AP5 was bath applied (Supplementary Fig. 7). Subsequently, similar experiments were carried out, but without injection of suprathreshold currents into postsynaptic neurons ('Non-associative', Fig. 5f). In these experiments, a lasting and statistically significant depression of the mean synaptic amplitude to 78 ± 10% was observed (Fig. 4g, 15–20-minute bin; n = 14 paired recordings; two-sided Wilcoxon signed-rank test, p = 0.042). Taken together, human synapses between pyramidal neurons in neocortical layers 2 & 3 exhibit differential plasticity, resulting in overall stabilization when associative activity occurs and lasting depression when non-associative activity occurs (between paradigm comparison: n = 41 paired recordings; two-sided Mann-Whitney U test, p = 0.0097; Fig. 5h).

## Discussion

Consolidation of explicit memories during sleep relies on precise coupling of spindle and ripple oscillations to SWA[18,31,35–37,39,41–44]. Impaired coupling is associated with weaker memory performance[36,38]. During coupled sleep oscillations, reactivation of neocortical memory traces works particularly well[31,33]. However, it is currently unknown what mechanisms mediate this efficient reactivation and how this promotes consolidation.

In-vivo UP and DOWN states during SWA are complex network phenomena involving an intricate balance of excitation and inhibition[21,24,25]. Somato-dendritic membrane potential de- and hyperpolarizations during UP and DOWN states passively spread along the axon and reach proximal presynaptic terminals (Fig. 1f)[46]. In this study, we found that such presynaptic membrane potential states modulate axonal AP-shape and hence synaptic reliability between human layer 2 & 3 pyramidal neurons. While UP-state-like depolarizations enhance presynaptic reliability, sequences of UP → DOWN → UP states further increase synaptic transmission, with a presumed maximum during DOWN-to-UP transitions (Fig. 4d–g). Besides AP-shape changes, other factors are known to affect presynaptic reliability during SWA. Notably, extracellular calcium concentration ($[Ca^{2+}]_o$) fluctuates during in-vivo SWA, de- and increasing during UP and DOWN states, respectively[64]. The decrease of $[Ca^{2+}]_o$ during UP states likely outweighs AP-broadening, as a net-decrease in synaptic reliability is observed with prolonged UP states in-vivo[64]. Conversely, the increased $[Ca^{2+}]_o$ during DOWN states boosts presynaptic reliability, adding to the effect of tuned APs observed during DOWN-to-UP transitions of UP → DOWN → UP sequences (Fig. 4d–g). Thus, the 'window-of-opportunity' for increased presynaptic potency in-vivo is presumably confined to the DOWN-to-UP transition during UP → DOWN → UP cycles, as APs are ideally tuned (Fig. 5i) and $[Ca^{2+}]_o$ is still high[73]. Since DOWN-to-UP transitions occur synchronously in numerous neurons during

sleep[20,74], transmission should be enhanced at multiple synapses simultaneously. Therefore, these transition periods likely represent time frames of broadly increased presynaptic potency in supragranular pyramidal neuron circuits. If neural activity, in particular activity triggered by sleep spindles and ripples, is coupled to a DOWN-to-UP transition, then it should lead to a more reliable activation of postsynaptic neurons, with a supra-linear relationship between the increase in synaptic strength and recruitment of APs (Fig. 5a). Reliable recruitment of postsynaptic APs during SWA-coupled presynaptic firing should result in increased co-firing of pre- and postsynaptic neurons (Fig. 5i). Utah-array recordings in humans during sleep have demonstrated such increased co-firing, which suggests that UP/DOWN-state-driven synaptic enhancement contributes to in-vivo neural dynamics[27]. Finally, we found that associated pre- and postsynaptic firing is a prerequisite for long-term stabilization of synaptic strength (Fig. 5i).

How could these mechanisms contribute to memory consolidation? We postulate that SWA modulates synaptic transmission in local neocortical circuits, thereby creating 'windows of opportunity' with increased presynaptic potency. These periods provide optimal conditions for spindles and ripples to achieve robust reactivation of neocortical ensembles, which store parts of memory traces[75–77]. This reactivation, in turn, serves as the basis for the long-term stabilization of ensembles due to associative plasticity. Thus, sleep spindles and ripples that are precisely coupled to SWA are privileged events implicated in promoting synaptic consolidation. Conversely, spindles that are not linked to SWA, termed 'uncoupled spindles'[78] or 'isolated spindles'[27,79], reach the cortex during a time period of comparably weaker presynaptic strength and are therefore less likely to induce co-firing and could lead to lasting depression of synapses (Fig. 5i). Such uncoupled spindles constitute a non-negligible fraction, with reported values ranging from ~25%[37,79] to ~45%[27] of all spindles. Whether uncoupled spindles are unwanted events or have a specific function (e.g., active forgetting[80]) is not resolved. However, the maturation of coupling during development[37] and its decline with older age[36], accompanied by an increase and deterioration of memory consolidation capabilities, suggests that uncoupled sleep oscillations are unintentional events, reflecting the inability of the brain to perfectly align neural activity at all times. In summary, our framework aligns well with the extensive evidence in humans, indicating that precise coupling of the faster sleep oscillations to SWA is essential for consolidation of declarative memories[18,31,35–37,39,41–44].

The synaptic plasticity rule described in this study, with associative stabilization on one hand and non-associative depression on the other hand (Fig. 5g), parallels in-vivo observations during sleep-like activity in mice[81] and is in good agreement with the 'down-selection' theory[82]. This theory points out that plasticity mechanisms during sleep should allow for protection of specific ensembles amidst global synaptic downscaling, which is necessary to maintain synaptic homeostasis and potentially allows for forgetting[2,12,80,82].

The role of SWA in creating 'windows of opportunity' for synaptic plasticity is expected to be particularly relevant for local neocortical synapses. Local ensembles of supragranular neocortical neurons contain parts of declarative memory traces in humans[75,76]. The entire trace of a memory is thought to include distributed ensembles across different brain regions, communicating via long-range connections[1,34,83]. Furthermore, other components of the neocortical circuit, such as inhibitory networks[84,85], interactions between cortical layers[81,86], dendritic[87] and postsynaptic mechanisms[88] are involved in computations during SWA and are believed to also contribute to memory consolidation. Thus, our results advance the concept of sleep-dependent consolidation by unraveling the role of SWA in boosting presynaptic strength and plasticity in local pyramidal neuron networks, which represent one important component of the brain-wide memory system.

Brain stimulation methods, both invasive and non-invasive, can be used to manipulate temporal coupling of sleep oscillations and were successfully applied to enhance memory performance[39–44]. From a synapse level point-of-view, our results indicate why precise coupling is beneficial and this could guide the development of improved stimulation strategies. According to our findings, neocortical pyramidal networks are tuned to maximal presynaptic strength after brief DOWN states (Fig. 4d–g). Therefore, we propose that aligning reactivation of engrams (potentially indicated by timing of hippocampal ripples[17,28–30]) to the neocortical DOWN-to-UP transition should be the best-educated approach for stimulation strategies in human subjects.

The neocortical tissue investigated in this study came from patients with refractory epilepsy or brain tumors. We were able to demonstrate modulation of synaptic transmission by subthreshold signals in tissue samples from both groups, suggesting that it is a more fundamental mechanism not related to a specific medical condition (Supplementary Fig. 3). While disease effects cannot be ruled out entirely, neurosurgical resections ultimately represent the only opportunity to investigate human synapses. Unraveling these basic synaptic mechanisms of the human brain is a fundamental step in advancing our understanding of sleep-dependent consolidation and could enable the development of improved therapeutic applications targeting memory performance in patients.

## Methods
### Human cortical tissue
All procedures strictly adhered to ethical requirements and were approved by independent ethics committees (Ethikkommission der Charité – Universitätsmedizin Berlin: EA2/111/14, EA2/064/22 in reference to EA4/206/20, EA2/086/20; Ethikkommission der Ärztekammer Westfalen-Lippe und der Westfälischen Wilhelms-Universität: 2020-517-f-S; Ethikkommission der Ärztekammer Hamburg: 2023-200674-BO-bet in reference to EA2/111/14). Human cortical tissue was obtained from patients undergoing neurosurgery for the treatment of drug-resistant epilepsy ($n = 33$) or brain tumors ($n = 12$). Tissue, that would otherwise be discarded, was used to perform the experiments in this study. All patients gave written consent for the scientific use of their resected tissue samples.

### Aggregate patient statistics
Cortical tissue samples were obtained from n = 45 patients. Samples were resected from temporal ($n = 38$ samples), frontal ($n = 5$) and parietal ($n = 2$) association cortices. The age of patients at the time of the surgery was $35.1 \pm 21.4$ years (mean ± s.d.; range: 3 – 77 years). Biological sex was self-reported by patients, or was reported by a legal guardian. N = 20 reported their sex to be female, while n = 25 reported it to be male. N = 12 of the patients had an intra-axial tumor. Of these, n = 7 had documented seizures in their patient history, whereas n = 5 had no documented seizures. The n = 33 patients without intra-axial tumors underwent surgery for the treatment of drug-resistant epilepsy (typically temporal lobe epilepsy due to hippocampal sclerosis).

### Acute brain slice preparation
After neurosurgical resection, tissue samples were immediately submerged in a sterile bottle containing ice-cold, carbogen-gassed (95/5% $O_2/CO_2$) sucrose-containing artificial cerebrospinal fluid (sACSF, in mM, 87 NaCl, 1.25 NaH$_2$PO$_4$, 2.5 KCl, 0.5 CaCl$_2$, 3 MgCl$_2$, 10 Glucose, 25 NaHCO$_3$, 75 Sucrose; filtered with a sterile 0.2 μm pore size filter). The bottle was sealed gas-tight and placed in a styrofoam box with ice for transport to the laboratory, which typically took <30 minutes. A vibratome (VT1200, Leica Biosystems) was utilized to cut the samples into 300-μm-thick acute brain slices. Slices were transferred into sterile storage containers filled with sACSF, which was heated to 34 °C for 30 minutes to support resealing of the cut membranes (recovery period). After recovery, the slices were kept in continuously carbogen-

gassed sACSF at room temperature (22–24 °C) until recordings were started. In a subset of experiments the slicing procedure was undertaken in a remote hospital and after recovery, slices were transported to our laboratory (~4 h trip) in a special transport container. In another small subset of experiments, the entire resected tissue piece was transported for ~2.5 h from a remote hospital to our laboratory in a gas-tight bottle containing ice-cold, carbogen-gassed sACSF.

## Electrophysiology

Patch-clamp recordings were performed under submerged conditions at 35 ± 2 °C. The bath chamber was perfused with ACSF solution (in mM, 125 NaCl, 1.25 NaH$_2$PO$_4$, 2.5 KCl, 2 CaCl$_2$, 1 MgCl$_2$, 10 glucose, and 25 NaHCO$_3$). For long-term plasticity experiments ('Associative' and 'Non-associative' paradigms) the concentration of CaCl$_2$ in the ACSF was reduced to 1.2 mM[72]. A subset of subthreshold modulation experiments was also performed at 1.2 mM extracellular calcium concentration (Supplementary Fig. 2). For certain experiments (see Main text), 100 nM DCG-IV (Tocris Bioscience) or 50 µM D-AP5 (Cayman Chemical) were added to the ACSF solution. Signals were recorded using MultiClamp 700B amplifiers (Axon Instruments) and digitized using a CED Power1401 board (Cambridge Electronic Design). Data acquisition was performed in Signal (Cambridge Electronic Design, version 6.06). Patch pipettes were pulled from borosilicate glass capillaries on a horizontal puller (P-97, Sutter Instrument Company). Standard pipettes (outer diameter: 2 mm; wall-thickness: 0.5 mm, Hilgenberg) were used for somatic recordings, resulting in access resistances of ~5-30 MΩ. To reduce wash-out of the cytosol during long-term plasticity experiments ('Associative' and 'Non-associative' paradigms), thick-walled pipettes were pulled to have small tip openings (wall-thickness: 0.7 mm; ~15–20 MΩ tip resistance) and were used for somatic recordings of presynaptic neurons (this typically resulted in access resistances >25 MΩ; plasticity experiments were excluded from analysis when the presynaptic access resistance was <25 MΩ). For axonal 'bleb' recordings (see below) thick-walled pipettes (wall-thickness: 0.7 mm; ~15–20 MΩ tip resistance) were used. Pipettes were filled with K-Gluconate intracellular solution (in mM, 130 K-gluconate, 2 MgCl$_2$, 0.2 EGTA, 10 Na$_2$-phosphocreatine, 2 Na$_2$ATP, 0.5 NaGTP, 10 HEPES, 0.1% Biocytin, 290–295 mOsm, pH adjusted to 7.2 with KOH). None of the membrane voltages reported in this study were corrected for the liquid junction potential, which we measured to be −15.2 ± 0.5 mV (mean ± s.d.) using our ACSF and pipette solution[89]. Cells were identified as pyramidal neurons based on (i) typical appearance in infrared-DIC video-microscopy, (ii) intrinsic properties (e.g., regular spiking, frequency adaptation, typical AP shape), (iii) excitatory output synapses and (iv) typical pyramidal neuron morphology (morphology of Biocytin filled neurons was examined post-hoc in a subset of neurons).

## Multineuron patch-clamp recordings

Multineuron patch-clamp recordings were performed as previously described[57]. In brief, up to ten somatic whole-cell patch-clamp recordings were established in a brain slice. All neurons were recorded in current-clamp mode. Data was low-pass filtered at 6 kHz and sampled at 20 kHz. Brief step-current injections (1–8 ms, 1–4 nA) were used to elicit APs. Typically, a short screening protocol was used to identify unitary connections. Afterward, the different stimulation paradigms reported in this study were performed.

## Axonal current-clamp recordings

In acute brain slices, cut axons form a structure at the slice surface termed 'bleb'[46]. These 'blebs' are accessible to patch-clamp recordings[46,47]. To establish a paired recording configuration of the soma and the axon, we first performed somatic whole-cell recordings using pipettes filled with intracellular solution containing Alexa Fluor 568 dye (Thermo Fisher Scientific). After ~5 minutes, the fluorescent

signal was used to identify the 'bleb' of the axon of a respective neuron. We then established a 'whole-bleb' recording using a thick-walled pipette. Signals recorded in this paired somato-axonal configuration in current-clamp mode were low-pass filtered at 30 kHz and sampled at a rate of 100 kHz. Somatic current injections were used to induce sub-threshold de- and hyperpolarizations and brief suprathreshold somatic current injections were used to elicit APs. The distance from the soma to the axonal recording site in the XY-plane was measured along the fluorescent signal of the Alexa Fluor 568 dye filled axon using cellSens Dimension software (Olympus). The Z-axis offset was accounted for using Pythagorean equation.

## Axonal voltage-clamp recordings

We performed voltage-clamp recordings in the 'whole-bleb' configuration (Fig. 2). The recording configuration was established as described in the section 'axonal current-clamp recordings'. Tetrodotoxin (0.5 µM, Hello Bio Ltd) and XE991[47] (10 µM, Tocris Bioscience) were added to the ACSF. The concentration of EGTA in the intracellular solution was increased to 10 mM. The sampling frequency was set to 50 kHz and the signal was filtered at 10 kHz. The holding voltage was −80 mV. A − P/4 protocol was used to correct for leak and capacitive currents. To establish voltage dependence of activation, the following protocol was used: 200 ms prepulses to −110 mV, followed by test pulses ranging from −90 to +70 mV in steps of 20 mV. To establish voltage dependence of inactivation, the following protocol was used: 5 s conditioning pulses ranging from −110 to 0 mV in steps of 10 mV, followed by test pulses to +40 mV. Chord-conductances were computed from peak currents using the formula $G = I_{peak} / (U_{test} − U_{rev})$, where $I_{peak}$ is the peak current in response to a test pulse, $U_{test}$ is the voltage during the test pulse and $U_{rev}$ is the reversal potential of the potassium current for the potassium concentrations used ($U_{rev} = −105$ mV). Conductances were normalized to the maximum conductance of each experiment ($G/Gmax$). Voltage-conductance relationships were fit with Boltzmann functions of the form $G/Gmax = bottom + (top − bottom) / (1 + exp((V1/2 − x) / slope))$, where $x$ is the voltage during test pulses or conditioning pulses, $V1/2$ is the membrane voltage at which half of the conductance was activated or inactivated, and $slope$ is the slope factor. To obtain time constants for inactivation and recovery from inactivation kinetics, 3 s long pulses from a holding voltage of −80 mV to +20 mV followed by brief pulses to +20 mV at variable intervals (100, 300, 1000 and 3000 ms, Fig. 2d) were used. The resulting currents were normalized and the inactivation-period as well as the recovery-period were separately fit with the sum of two exponential functions. A custom-made pressure system[57] was employed for puff-application of Dendrotoxin-I (500 nM in HEPES-ACSF, Alomone Labs). Puff-application was directed at the 'bleb' and the segment of the axon entering the 'bleb'.

## Data analysis

Electrophysiology traces were analyzed off-line using custom-written Matlab (MathWorks, R2021a) scripts. Data of paired whole-cell recordings of synaptically connected pyramidal neurons were manually curated to remove single trials where (i) current injection failed to trigger presynaptic AP, (ii) the pre- or postsynaptic neuron depolarized spontaneously or (iii) spontaneous postsynaptic activity interfered with measurement of triggered EPSP. To measure the amplitude of EPSPs, the steepest rise of the presynaptic AP was detected. The postsynaptic baseline was then defined as the mean of a 5 ms segment in the time window before the presynaptic AP. The position of the baseline segment was adjusted if it was affected by a stimulation artifact. The peak of the postsynaptic trace in a 25 ms window after the presynaptic AP was then detected. We took the mean of a 0.5 ms segment surrounding this peak and subtracted the baseline to get the EPSP amplitude. To obtain the length constant for the attenuation of passively spreading signals in the axon (Fig. 1f), step currents were

injected into the soma of pyramidal neurons to cause a subthreshold depolarization, which spread to the axonal recording sites. The steady state amplitude of these depolarizations was measured at the soma and the axon, and a ratio was calculated (steady state amplitude axon / steady state amplitude soma). A mono-exponential decay function was fit to the ratios plotted over the distance between the soma and the axonal recording site (Fig. 1f) and a decay constant ($\lambda$) was computed. The axonal AP amplitude was measured from the resting membrane potential of the axon to the peak value of the AP. The half-duration was defined as the duration of the AP at half amplitude. The axonal AP overshoot was measured from 0 mV to the peak value of the AP. Recordings were excluded from axonal AP shape analyses when the axonal recording site was <50 µm from the soma ($n$ = 2). After raw trace analysis, source data tables were exported for statistical analyses and graph plotting in RStudio (Posit PBC, R version 4.2.1). Figures were assembled in CorelDraw (Alludo, 2019).

### Statistical analysis

The Wilcoxon signed-rank test was used to assess the statistical significance of changes in synaptic amplitude, synaptic reliability and axonal AP shape for all subthreshold modulation paradigms. A p-value for the Pearson correlation coefficient was computed based on the t-statistic. The Fisher exact test was used to test the significance of the difference in the number of triggered postsynaptic APs in response to 'Control' vs. 'Depol→Hyperpol' paradigm in convergent motifs (Fig. 5a). The Mann-Whitney $U$ test was used to assess the significance of difference between effects of opposing plasticity induction paradigms (e.g., 'Associative' vs. 'Non-associative', Fig. 5h). The Wilcoxon signed-rank test was used to test the significance of long-term synaptic changes in response to a specific plasticity paradigm across multiple experiments. For within-experiment analysis of long-term plasticity, ARIMA models were used (Supplementary Fig. 6). Two models with identical parameters were fit to the time series data (amplitude of synapse over time), except that one model accounted for plasticity induction with a step function. Akaike information criterion was used to assess whether the model that accounted for induction fit the data better. All tests were two-sided.

### Reporting summary

Further information on research design is available in the Nature Portfolio Reporting Summary linked to this article.

## Data availability

Source data are provided with this paper. Source data tables and code to reproduce the results and visualizations are also deposited in a public figshare repository (https://doi.org/10.6084/m9.figshare.24793218.v1). Raw electrophysiological traces files from human samples cannot be made publicly available due to German and European privacy law restrictions. Source data are provided with this paper.

## Code availability

The code used to compute the results and statistics, as well as generate visualizations is deposited in a public figshare repository (https://doi.org/10.6084/m9.figshare.24793218.v1).

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

## Acknowledgements

We thank A. Wilke for her excellent technical assistance. We thank N. Maier and D. Owald for comments on earlier versions of the manuscript. The resources for this project were provided by institutional funding of the Institute of Neurophysiology, Charité – Universitätsmedizin Berlin, corporate member of Freie Universität Berlin and Humboldt Universität zu Berlin (JG). This research was funded by the Deutsche Forschungsgemeinschaft (DFG, German Research Foundation) under Germany´s Excellence Strategy – EXC-2049 – 390688087.

## Author contributions

F.M., H.A., and J.G. conceptualized the project. F.M. performed data acquisition, data analysis and data visualization. S.G. imaged & reconstructed Biocytin-stained neurons. T.K., R.X., J.O., K.F., T.S., U.W.T., A.M.K., M.H., M.S. screened patients, informed patients to get consent and contributed to neurosurgical tissue acquisition. P.F., H.A., and J.G. coordinated cooperations and obtained ethical approvals. J.G. organized resources and institutional funding. F.M., H.A., and J.G. wrote the original draft of the manuscript. All authors reviewed and edited the manuscript.

## Funding

## Competing interests

The authors declare no competing interests.
