## [Peer Review File · Nature Communications]

REVIEWER COMMENTS

Reviewer #1 (Remarks to the Author):

The manuscript by Mittermaier and co-workers is aimed at exploring the role of presynaptic membrane potential states on synaptic transmission in human neocortical L2/3 neurons. This question enters into the mechanisms underlying sleep-associated memory consolidation. Using paired whole-cell recordings from L2/3 pyramidal neurons in tissues samples obtained from patients who underwent neurosurgery for the treatment of epilepsy or brain tumors, the authors show that slow membrane oscillations known as UP and DOWN states enhance synaptic transmission. Using whole-cell recording from the cut axon, they show that this synaptic enhancement is due to the broadening of the action potential. Next, they show that sequences of presynaptic depolarization followed by a brief hyperpolarization constitute the optimal stimulus to enhance synaptic transmission as this sequence of events not only broadens the axonal AP but also increase its amplitude. Finally, using triple recordings from two presynaptic neurons and one postsynaptic neuron, the authors show that synchronous activation of the presynaptic neurons during UP DOWN state transitions increase the EPSP-evoked firing in the postsynaptic cell. This protocol leads to long-lasting synaptic enhancement of synaptic transmission whereas presynaptic firing alone induces long-lasting depression of synaptic transmission. The authors conclude that UP DOWN state transitions that occur during sleep lead to synaptic enhancement in human neurons.

This paper is technically very impressive, well written and the quality of the data is outstanding. In addition, the general message carried by this paper is really new and original as it shows that short-term facilitation of synaptic transmission in human neurons may induce long-lasting enhancement when postsynaptic spiking occurs. Nevertheless, a few minor points deserve specific attention.

1. Does presynaptic hyperpolarization alone followed by spiking is sufficient to induce synaptic enhancement in human neurons as it has been previously shown in rodent synapses? Perhaps, the authors have already obtained these data.
2. What is the precise sequence of the experiment illustrated in Fig. 4a? The weak postsynaptic firing produced without UP / DOWN presynaptic fluctuation may influence the subsequent postsynaptic firing when the UP / DOWN configuration is tested.
3. The post-synaptic spike number evoked by trains of presynaptic APs may vary from synapse to synapse. What is the incidence of AP number on the magnitude of short-term (first 2 min) and long-term potentiation (10-15 min)? A simple analysis of the existing data would be sufficient.
4. Synaptic potentiation is generally associated with postsynaptic firing enhancement. Did the authors test this in human neurons?
5. What is the frequency of uncoupled spindle during sleep? A small paragraph on this would be helpful in the discussion.

Reviewer #2 (Remarks to the Author):

This manuscript explores how changes in subthreshold membrane potentials modifies synaptic potentials and neuronal output in human layer 2/3 neocortical pyramidal neurons. The authors made brain slices from tissue obtained from patients with refractory epilepsy or brain tumors and performed whole-cell patch clamp recordings from synaptically connected pyramidal neurons or soma and axon-blebs. Finally, the authors' results showed that NMDA-receptor mediated synaptic plasticity could be induced in these human neurons.

Overall, the findings are very interesting, particularly given the lack of knowledge on human neuron physiology. The findings suggest that subthreshold depolarizations modulate K⁺ channels (presumably the A-type or KV1.1 K⁺ channels) to enhance pre-synaptic action potential widths and increase reliability of synaptic transmission under these conditions. Whilst the authors point to this mechanism in the text, the authors did not further investigate this. It would have been interesting if the authors had investigated this further as this would provide a cellular mechanism by which these effects might occur in humans.

The authors also suggest that their experimental paradigms mimic UP and DOWN states and this may lead to a 'window' of enhanced synaptic plasticity in these neurons. Whilst these experiments are interesting, it is likely that the hyperpolarization increased the availability of Na⁺ channels and thereby boosted action potentials during the subsequent depolarization. Again, whilst this is interesting, there remain a number of questions that remain to be answered.

Concerns

1. Most of the human tissue samples (32/36) collected by the authors were from patients who had experienced seizures (Extended data table 1). Further, there is a significant variation in patient ages ranging from 3 years old to 77 years old. Whilst the authors claim there are no differences in their findings from samples obtained from patients with a history of seizures compared with no seizures, the sample size for those that had no seizures is too low to make such a comparison. Thus, it is important to increase the number of samples obtained from patients with no seizure history to make such a comparison. This is particularly important given that ion channel biophysical properties and densities are often altered with induction of epilepsy. Further, it would be worthwhile

to determine if there is a developmental effect on their findings (i.e. do findings from samples obtained from children differ from findings obtained from adults).

2. There is little information available about the amount of depolarization required for the effects on observed synaptic transmission (Fig 1, 2, 3). From the figures, it seems that at least 20 mV depolarization was applied pre-synaptically. If so, this is very large. Would the authors find similar findings if 5 mV depolarizations, which perhaps would be more typical, are applied?

3. Whilst the authors systematically investigated the duration of depolarization required to obtain the effects, did the authors also investigate whether the duration of hyperpolarization in the sequence of depolarization + hyperpolarization + depolarisation steps (Fig 3, Fig 4) is critical for the increase in synaptic strength and plasticity?

4. Did the authors attempt to determine if a decrease in KV1 channel function is likely to account for the increased action potential duration induced by the depolarization, particularly as there are reports (e.g. Vivekananda et al., PNAS, 2017) that suggest that KV1.1 subunits may not be necessary for subthreshold modulation of spike width in rodents?

5. Did the authors investigate as to why hyperpolarization did not reverse the effects on K⁺ channels induced by depolarization. Some insight into this would enhance the significance of their findings.

Response to Reviewer 1

Reviewer 1:

The manuscript by Mittermaier and co-workers is aimed at exploring the role of presynaptic membrane potential states on synaptic transmission in human neocortical L2/3 neurons. This question enters into the mechanisms underlying sleep-associated memory consolidation. Using paired whole-cell recordings from L2/3 pyramidal neurons in tissues samples obtained from patients who underwent neurosurgery for the treatment of epilepsy or brain tumors, the authors show that slow membrane oscillations known as UP and DOWN states enhance synaptic transmission. Using whole-cell recording from the cut axon, they show that this synaptic enhancement is due to the broadening of the action potential. Next, they show that sequences of presynaptic depolarization followed by a brief hyperpolarization constitute the optimal stimulus to enhance synaptic transmission as this sequence of events not only broadens the axonal AP but also increase its amplitude. Finally, using triple recordings from two presynaptic neurons and one postsynaptic neuron, the authors show that synchronous activation of the presynaptic neurons during UP DOWN state transitions increase the EPSP-evoked firing in the postsynaptic cell. This protocol leads to long-lasting synaptic enhancement of synaptic transmission whereas presynaptic firing alone induces long-lasting depression of synaptic transmission. The authors conclude that UP DOWN state transitions that occur during sleep lead to synaptic enhancement in human neurons.

This paper is technically very impressive, well written and the quality of the data is outstanding. In addition, the general message carried by this paper is really new and original as it shows that short-term facilitation of synaptic transmission in human neurons may induce long-lasting enhancement when postsynaptic spiking occurs. Nevertheless, a few minor points deserve specific attention.

- We thank the Reviewer for their positive assessment of our study. We fully acknowledge the points raised. We will address them in the point-by-point response below and thank the Reviewer for their constructive feedback. We used the following color-code: grey text is original version of the manuscript, red text is new, purple strikethrough-text was deleted from original version.

1. Does presynaptic hyperpolarization alone followed by spiking is sufficient to induce synaptic enhancement in human neurons as it has been previously shown in rodent synapses? Perhaps, the authors have already obtained these data.

- We appreciate this important question. To address whether isolated hyperpolarizations modulate axonal APs and affect presynaptic release, we performed paired somato-axonal recordings, as well as paired somatic recordings of synaptically connected pyramidal neurons in layers 2 & 3 of the human neocortex. Negative step currents (200 ms) were injected into the soma to induce hyperpolarizations, which passively spread into the axons. Following such hyperpolarizations, APs were elicited by injecting brief suprathreshold current pulses into the soma ('Hyperpol' condition).

Stimulation paradigm:

- The average resting membrane potential of recorded axons was -74.4 ± 5 mV (mean \pm s.d., $n = 10$ axons) but axonal resting could be as negative as -81 mV (not corrected for liquid junction potential, which we measured to be -15.2 ± 0.5 mV, mean \pm s.d.¹). In those axons that had more depolarized resting membrane potentials (> -77 mV, 5/10 axons), the overshoot amplitude of axonal APs increased

during the 'Hyperpol' condition relative to 'Control'. The overshoot amplitude did not increase in axons that had resting membrane potentials more negative than -77 mV.

- This suggests that hyperpolarizations lead to a recovery of voltage gated sodium channels (Na_v) in more depolarized axons, whereas nearly all Na_v s are available for activation in axons with more negative resting membrane potentials.
- In line with the axonal AP data, modulation of synaptic transmission was not as pronounced as it was in response to the 'Depol→Hyperpol' condition. Nevertheless, we found a statistically significant increase in average EPSP amplitude of 12% ($n=21$ unitary synapses, Wilcoxon signed-rank test, $p=0.012$).

- We combined the figure panels shown above into Supplementary Fig. 5 and added the following text to our manuscript:
 - "We also studied the effect of isolated hyperpolarizations (without preceding depolarizations, Supplementary Fig. 5). Isolated 200 ms long hyperpolarizations increased AP amplitudes only in those axons that had resting membrane potentials more depolarized than -77 mV (5/10 axons, Supplementary Fig. 5a-c), suggesting that these axons contain inactivated Na_v s at resting. Similar to what has been found in rodents⁵⁸, isolated hyperpolarizations enhanced synaptic strength by $12 \pm 4\%$ (mean \pm s.e.m., $n=21$ unitary synapses, $p < 0.05$, Supplementary Fig. 5d-e). Taken together, both de- and hyperpolarizations can separately modulate axonal AP shape and synaptic transmission, but sequences of de- and brief hyperpolarizations represent the ideal stimulus to boost synaptic strength."

2. What is the precise sequence of the experiment illustrated in Fig. 4a? The weak postsynaptic firing produced without UP / DOWN presynaptic fluctuation may influence the subsequent postsynaptic firing when the UP / DOWN configuration is tested.

- We acknowledge that Fig. 4a doesn't allow to infer the exact design of the stimulation protocol, even though this information is important for the reason mentioned by the Reviewer. When we conceptualized these experiments, we had the same concern as the Reviewer: An extended period of weak firing in the postsynaptic neuron could affect the neurons response to synaptic input. To mitigate this problem, an interleaved stimulation paradigm was used. The 'Control' and 'Depol→Hyperpol' conditions were induced in an alternating fashion, separated by 5 s long intervals.

- To allow readers to better understand how the experiments were conducted, we updated the figure panel (see revised manuscript: Fig. 5a) and thank the Reviewer for bringing this shortcoming to our attention.

3. The post-synaptic spike number evoked by trains of presynaptic APs may vary from synapse to synapse. What is the incidence of AP number on the magnitude of short-term (first 2 min) and long-term potentiation (10-15 min)? A simple analysis of the existing data would be sufficient.

- To address this valid and interesting question, we reanalyzed our data. For each of the 'Pre drive post' experiments (Fig. 5b-c in the revised manuscript shows this experimental paradigm), we counted the APs that were evoked in the postsynaptic pyramidal neuron via the convergent synaptic motif during the induction period. In a scatter plot, the number of evoked postsynaptic APs was plotted on the x-axis and the magnitude of synaptic potentiation at the end of the 2.5-minute post-induction time-window (80-160 seconds after the end of induction, Fig. 5d-e) was plotted on the y-axis. The 'Pre fail to drive post' experiments were included in the graph (blue data points). By design, no postsynaptic APs were evoked during the induction periods of these 'Pre fail to drive post' experiments (except for n=2 experiments, where 1 and 2 postsynaptic APs occurred unintentionally). We calculated the correlation coefficient (Pearsons R = 0.52) and computed a p-value based on the t-statistic (p = 0.03).

- Next, the associative plasticity experiments (see revised manuscript: Fig 5f-h) were reanalyzed. These experiments were performed in pairs of synaptically connected pyramidal neurons and postsynaptic APs were evoked by a combination of synaptic input and postsynaptic current injection, as illustrated in Fig 5f. The evoked postsynaptic APs were counted and the relationship between the number of postsynaptic APs and the magnitude of long-term synaptic plasticity (10-15 min post-induction) was plotted analogously to the graph above. The correlation coefficient was computed to be $R = 0.3$ ($p = 0.06$).
- Next, only those synapses that had a clear within-experiment long-term potentiation or depression effect (as determined by ARIMA time series analysis, see Supplementary Fig. 6) were included. In this subgroup of synapses, a significant positive correlation between the number of induced postsynaptic APs and the long-term plasticity effect was detectable ($R = 0.6$, p -value = 0.007).

- Taken together, our data shows a trend towards stronger plasticity induction when the postsynaptic neuron fires more APs, which are temporally associated with the presynaptic input. We combined the figure panels shown above into Supplementary Fig. 8.

4. Synaptic potentiation is generally associated with postsynaptic firing enhancement. Did the authors test this in human neurons?

- The term 'synaptic potentiation' could refer to both short- as well as long-term potentiation. Since we had demonstrated postsynaptic firing enhancement as a result of short-term potentiation in the initial manuscript (see revised manuscript: Fig. 5a; formerly Fig. 4a), we infer that the Reviewer is asking about long-term potentiation.
- Unfortunately, we did not test whether long-term potentiation of human synapses is associated with increased postsynaptic AP firing, and we argue that our methodological repertoire is not well suited to address this question.
- For the investigation of long-term plasticity (timescale of >10 minutes), we used paired whole-cell recordings of synaptically connected pyramidal neurons. This approach allowed us to study unitary synaptic connections with known identity of pre- and postsynaptic neurons (in this study: layer 2&3 pyramidal neurons). Alternative techniques in human tissue, such as tract stimulation approaches, do

not have this single synapse resolution. However single unitary synaptic connections between cortical pyramidal neurons typically generate comparatively small EPSP amplitudes (0.1 - 3 mV). Therefore, single synapses are usually not capable of triggering postsynaptic APs. Even if a synapse would undergo strong (e.g. 80%) potentiation due to long-term plasticity mechanisms, this constraint would most likely still remain. Multiple synapses that converge onto a postsynaptic neuron are necessary to reach the AP-threshold. Tract stimulation, for instance, can recruit multiple presynaptic axons and generate large compound PSPs that can trigger postsynaptic APs. However, this comes at the expense of not knowing the identity of the presynaptic neurons.

- Our multineuron patch-clamp approach allowed us to identify convergent motifs of unitary synaptic connections and these recordings were used to trigger postsynaptic APs in the experiments shown in Fig. 5a & b (formerly Fig. 4 a & b). However, these recordings are technically extremely challenging and did not allow for recording times on the scale >10 minutes.
- Taken together, unitary synapses are usually not strong enough to trigger postsynaptic APs and multineuron recordings are typically not stable enough to record for longer time periods.
- But given our results, future studies, in particular in-vivo unit recording studies in the human neocortex^{2,3}, could be designed to address this question. For instance, one could ask whether an identified regular spiking unit changes its firing frequency after an episode of slow wave sleep, during which the unit was recruited into co-firing with neighboring neurons during UP/DOWN-state coupled sleep spindles.

5. *What is the frequency of uncoupled spindle during sleep? A small paragraph on this would be helpful in the discussion.*

- We appreciate the Reviewers' valuable question and agree that adding a paragraph, that discusses this aspect, would benefit readers and enhance the clarity of our manuscript. The frequency of occurrence of 'uncoupled spindles' (also referred to as 'isolated spindles') during sleep can be inferred from the spindle frequency of occurrence (spindles/minute) and the percentage of uncoupled spindles. Estimates of these values have been reported in several studies. It is worth mentioning that these numbers depend on the recording technique (e.g. surface EEG vs. ECoG or iEEG), the exact algorithms used to detect oscillation events and the approach that is being used to determine coupling. Therefore, reported values in the literature of human sleep vary.
- For instance, Dickey et al.² report: "*We found that 45.11% of spindles did not begin within ± 1000 ms of downstates or upstates (henceforth referred to as "isolated" spindles) (...)*". This study, which was conducted on epilepsy patients, reports a mean spindle density of ~11 spindles/minute during NREM sleep.
- A different study on epilepsy patients undergoing presurgical invasive EEG diagnostics reports the percentage of isolated spindles to be 35% of all spindles⁴. They define isolated spindles as spindles without a slow wave within ± 1.5 s. In this study the mean spindle density was lower and amounted to ~2 spindles/minute.
- In another study⁵ on healthy adolescent participants (16 ± 0.9 years years), it was found that 26.8% of sleep spindles had no UP/DOWN states in a ± 2.5 s window around the spindle-peak. For one individual in their dataset, the data on spindle density is publicly available and was 12.03 spindles/minute.
- Further studies report uncoupled spindles, but often do not state their exact percentage or density (e.g. References⁶⁻⁸).
- Taken together, the emerging picture is that approximately 25-45% of spindles are not coupled to slow wave activity. If we would calculate with a density of 10 spindles/minute, then this would give us a frequency of ~3 uncoupled spindles per minute. We added the following paragraph to our discussion:
 - "How could these mechanisms contribute to memory consolidation? We postulate that SWA modulates synaptic transmission in local neocortical circuits, thereby creating 'windows of opportunity' with increased synaptic potency. These windows provide optimal conditions for spindles and ripples to achieve a robust reactivation of neocortical ensembles, which store parts of memory traces^{66,67}. This reactivation, in turn, serves as the basis for the long-term stabilization of ensembles due to associative plasticity. **Thus, sleep spindles and ripples that are precisely coupled to SWA are privileged events implicated in promoting synaptic consolidation. Conversely, spindles that are not linked to SWA, termed 'uncoupled spindles'⁶⁸ or 'isolated spindles'^{36,69}, are**

less likely to induce co-firing and could therefore lead to long-term depression of synapses (Fig. 5i). Such uncoupled spindles constitute a non-negligible fraction, with reported values ranging from ~25%^{10,69} to ~45%³⁶ of all spindles. Whether uncoupled spindles are unwanted events or have a specific function (e.g., active forgetting⁷⁰) is not resolved. However, the maturation of coupling during development¹⁰ and its decline with older age⁹, accompanied by an increase and deterioration of memory consolidation capabilities, suggests that uncoupled sleep oscillations are unintentional events, reflecting the inability of the brain to perfectly align neural activity at all times.”

- Furthermore, we added typical oscillation frequencies of the cardinal NREM sleep rhythms to our introduction:
 - “This brain state gives rise to characteristic oscillation patterns in the electroencephalogram, including slow waves (~0.5-4 Hz), sleep spindles (~10-16 Hz) and hippocampal ripple oscillations (~80-120 Hz in humans).”

Response to Reviewer 2

Reviewer 2:

This manuscript explores how changes in subthreshold membrane potentials modifies synaptic potentials and neuronal output in human layer 2/3 neocortical pyramidal neurons. The authors made brain slices from tissue obtained from patients with refractory epilepsy or brain tumors and performed whole-cell patch clamp recordings from synaptically connected pyramidal neurons or soma and axon-blebs. Finally, the authors' results showed that NMDA-receptor mediated synaptic plasticity could be induced in these human neurons.

Overall, the findings are very interesting, particularly given the lack of knowledge on human neuron physiology. The findings suggest that subthreshold depolarizations modulate K⁺ channels (presumably the A-type or KV1.1 K⁺ channels) to enhance pre-synaptic action potential widths and increase reliability of synaptic transmission under these conditions. Whilst the authors point to this mechanism in the text, the authors did not further investigate this. It would have been interesting if the authors had investigated this further as this would provide a cellular mechanism by which these effects might occur in humans.

The authors also suggest that their experimental paradigms mimic UP and DOWN states and this may lead to a 'window' of enhanced synaptic plasticity in these neurons. Whilst these experiments are interesting, it is likely that the hyperpolarization increased the availability of Na⁺ channels and thereby boosted action potentials during the subsequent depolarization. Again, whilst this is interesting, there remain a number of questions that remain to be answered.

- We appreciate the Reviewers' feedback and thank the Reviewer for contributing their time. We acknowledge the concerns raised and will thoroughly address them in the point-by-point response below. We used the following color-code: grey text is original version of the manuscript, red text is new, purple strikethrough-text was deleted from original version.

Concerns

1. Most of the human tissue samples (32/36) collected by the authors were from patients who had experienced seizures (Extended data table 1). (...) Whilst the authors claim there are no differences in their findings from samples obtained from patients with a history of seizures compared with no seizures, the sample size for those that had no seizures is too low to make such a comparison. Thus, it is important to increase the number of samples obtained from patients with no seizure history to make such a comparison. This is particularly important given that ion channel biophysical properties and densities are often altered with induction of epilepsy.

- The Reviewer raises an important point: The tissue samples investigated in this study were obtained from patients undergoing neurosurgery and could thus have been affected by the disease background. We are aware of this inherent limitation and had already stated this in the discussion of our initial manuscript:
 - "While disease effects cannot be ruled out entirely, neurosurgical resections ultimately represent the only opportunity to investigate human synapses."
- For the initial version of this manuscript, we had collected samples from patients with different medical conditions in an attempt to counter this limitation, by ruling out that the observed effects are specific to one particular disease entity.
- The broad groups (refractory epilepsy/tumor; seizures/no seizures) were not balanced and reflect the availability of tissue samples. Samples from anterior temporal lobe resections, which are performed to treat patients with certain forms of temporal lobe epilepsy, are the most abundant samples that we can study (the cortical samples that are resected during this procedure are comparatively large, yielding more brain slices). Less often, we can obtain samples of sufficient size and without considerable damage (e.g. compression of the cortex by the tumor) from patients undergoing neurosurgical resection of brain tumors. Often, patients with brain tumors have symptomatic seizures. Only occasionally (4/36 in our dataset before the revision) do we receive tissue samples from patients that have no documented seizure in their medical history.

- It is important to clarify that, in the initial manuscript, we did not claim that “(...) there are no differences in their findings from samples obtained from patients with a history of seizures compared with no seizures (...)”. We merely made the qualitative statement, that the modulation of synaptic strength by subthreshold depolarizations can be observed in samples obtained from both tumor- as well as epilepsy patients:
 - “The neocortical tissue investigated in this study came from patients with refractory epilepsy or brain tumors. We were able to demonstrate modulation of synaptic transmission by subthreshold signals in tissue samples from both groups, suggesting that it is a more fundamental mechanism not related to a specific medical condition.”
- However, we agree that it would be advantageous to increase the number of samples from patients without a history of seizures. In the ~3-month timeframe for the revision of this manuscript (suggested initially by the editor), we received samples from n=1 patient without seizures. In line with previous observations, the synapses of this patient (n=4 unitary synapses) displayed modulation by presynaptic subthreshold depolarizations. For the ‘Depol’ stimulation paradigm (as in Fig. 1) the dataset now contains a total of n=12 synapses from n=5 patients without any documented seizures. For comparison, our dataset contains n=13 synapses from n=4 patients who underwent surgery for removal of a brain tumor, who had experienced seizures prior to their surgery. Furthermore, our dataset contains n=51 synapses from n=7 patients without brain tumors, who underwent surgery for the treatment of refractory epilepsy.

- First, we tested whether there is a statistically significant difference between the mean synaptic facilitation by presynaptic subthreshold depolarizations of synapses from patients with- and without seizures. We did not reject the H_0 of a *Mann-Whitney U* test ($p=0.29$) and conclude that the mean effects in the two groups are not significantly different.

- To address quantitatively whether synaptic modulation-effects from patients without seizures are ‘practically equivalent’ to those of patients with seizures, we performed an equivalence test (*TOST procedure*, *TOSTER*-package in R). The H_0 of this test was that the mean synaptic facilitation by presynaptic subthreshold depolarizations is not equivalent between the two groups. ‘Practical equivalence’ was defined as any difference in mean synaptic enhancement that is smaller than the boundary of $\pm 0.8 \cdot s.d.$ (*s.d.* refers to the standard deviation of the pooled data; Cohen⁹ suggested to

only consider effect sizes $>0.8*s.d.$ as large). We rejected the H_0 at a significance level of 0.05 and conclude that synaptic facilitation is 'practically equivalent' between patients with- and without seizures.

- We want to emphasize that, in comparison to peer-reviewed studies investigating human cortical tissue (e.g. References¹⁰⁻¹²), our study meets the established standard when it comes to disease-diversity of human tissue samples (refractory epilepsy & tumor resections, w and w/o history of seizures), reporting seizure history (we include an entire supplementary figure as well as a statement in the discussion to address this issue) and sample size (in total >200 unitary synaptic connections across all experimental paradigms, which is only outmatched by our own recent study: Peng et al., *Science* 2024¹³).
- While we can't force a larger sample size of seizure-free patients in a foreseeable amount of time, due to obvious ethical considerations, we argue that we openly communicate the data with regard to the disease background (Supplementary Fig. 3) and will make source data tables available to allow readers to properly interpret the data and draw an informed conclusion.

1.1. Further, there is a significant variation in patient ages ranging from 3 years old to 77 years old. (...)

Further, it would be worthwhile to determine if there is a developmental effect on their findings (i.e. do findings from samples obtained from children differ from findings obtained from adults).

- To determine if there is evidence for developmental effects, we reanalyzed our dataset (including data that was acquired during the revision). Firstly, the enhancement of synaptic transmission by presynaptic subthreshold depolarizations ('Depol' condition) was assessed with respect to the age of the patients that the samples were obtained from. In total, this effect was studied in $n=76$ synapses from $n=16$ patients. There was no significant correlation between the age of patients and the magnitude of synaptic enhancement (Pearson correlation coefficient $R = -0.05$, t-statistic based $p = 0.6$).

- Next, the broadening of axonal APs by subthreshold depolarizations was reanalyzed with respect to the age of the patients. The dataset contained $n=15$ paired somato-axonal recordings from $n=9$ patients. No significant correlation was observed between the age and the magnitude of axonal AP broadening in response to subthreshold depolarizations ($R = -0.06$, $p = 0.8$).

- Next, the effect of sequences of de- and hyperpolarizations ('Depol→Hyperpol' condition) on synaptic transmission was reanalyzed. The dataset contained n=21 synapses from n=8 patients that were investigated using this experimental paradigm. No significant correlation was found ($R = -0.25$, $p = 0.3$).

- Taken together, there is no evidence for developmental effects. We combined the panels shown above into Supplementary Fig. 9.

2. There is little information available about the amount of depolarization required for the effects on observed synaptic transmission (Fig 1, 2, 3). From the figures, it seems that at least 20 mV depolarization was applied pre-synaptically. If so, this is very large. Would the authors find similar findings if 5 mV depolarizations, which perhaps would be more typical, are applied?

- In our experimental paradigm the subthreshold depolarizations were tuned so that they narrowly fell short of triggering APs (i.e. the membrane at the soma was depolarized to just below the AP threshold voltage). As pyramidal neurons had variable resting membrane potentials, this procedure naturally led to a range of depolarization amplitudes from 10 to 26 mV.
- The Reviewer specifically asked whether an effect could be observed if 5 mV depolarizations are applied. From a theoretical perspective, a small depolarization should also induce inactivation of a fraction of voltage gated potassium channels (see response to point 4), provided that the resting membrane potential is positioned in the descending part of the inactivation-curve. However, the signal-to-noise ratio of cortical synapses is inherently low, making it difficult to detect small effects reliably. Nevertheless, we performed a set of experiments (n=7 unitary synapses) to address the Reviewers' comment. To make these new experiments more comparable to the original dataset, we used a continuous current injection to depolarize the presynaptic neuron to ~ -65 mV (measured at the soma). From this membrane potential of -65 mV an additional step current was injected to cause subthreshold depolarizations with low amplitudes (3-6 mV). APs were then elicited by brief suprathreshold currents either from -65 mV ('Control') or following the low amplitude depolarizations ('Depol').
- We then analyzed the entire dataset to test whether there is a relationship between the amplitudes of the induced depolarizations and the change in synaptic strength from 'Control' to 'Depol' condition. As would be expected from theoretical considerations, we observed a positive correlation ($R=0.19$, $p=0.09$). For a more intuitive interpretation we arbitrarily binned our dataset into five groups depending on the subthreshold depolarization amplitudes (5 mV bins).

- We created a new supplementary figure and added the following sentence to our results text:
 - “In addition to sufficient duration, depolarization amplitudes had to be >10 mV (measured at the soma) to reliably cause enhancement of synaptic transmission (Supplementary Fig. 1).”
- To establish whether depolarization amplitudes >10 mV are reasonable in the context of slow wave sleep, we revisited the literature on *in-vivo* membrane potential UP/DOWN states. We focused on mammalian species, cortical neurons and natural slow wave sleep (as opposed to slow oscillations induced by anesthesia). We found that *in-vivo* studies typically report amplitudes between 10 to 20 mV, measured from the hyperpolarized DOWN-state to the depolarized UP-state at the neuron soma (e.g. References¹⁴⁻¹⁶). To quote seminal work by Steriade, Timofeev & Grenier¹⁴: “Recordings of all electrophysiologically identified cortical cell types across the whole sleep-waking cycle demonstrated that the SWS state was distinguished from both waking and REM sleep by the presence of cyclic, long-lasting (0.3–0.5 s), high-amplitude (8–20 mV) hyperpolarizations (...)”. And in subsequent work by Chauvette, Volgushev & Timofeev¹⁵: “(...) somatic membrane potential in layer V large pyramids during active states is depolarized by up to 15–20 mV relative to silent states”.
- To our knowledge, there are no published *in-vivo* intracellular recordings from human neurons. However, we identified one study in non-human primates, that reports membrane potential oscillations under anesthesia. For one exemplary cortical neuron, the authors of this study show a bimodal distribution of the membrane potential (reflecting UP/DOWN states) with one mode at –63 mV and the other mode at –79 mV, indicating an UP-state amplitude of ~16 mV.
- While there remains uncertainty about the UP-state amplitude in human cortical neurons, we argue that we find an effect on synaptic transmission by inducing subthreshold depolarizations that are roughly in the expected range of *in-vivo* UP-states.

3. Whilst the authors systematically investigated the duration of depolarization required to obtain the effects, did the authors also investigate whether the duration of hyperpolarization in the sequence of depolarization + hyperpolarization + depolarisation steps (Fig 3, Fig 4) is critical for the increase in synaptic strength and plasticity?

- We appreciate this valuable and important question. We performed additional experiments to explore this aspect. In the paired somato-axonal recording configuration we induced 800 ms long depolarizations followed by hyperpolarizations of varying duration (50 ms, 200 ms and 1 s).

- 50 ms long hyperpolarizations were sufficient to restore the axonal AP amplitude. This is in line with the fast recovery from inactivation of Na_v channels (in nucleated patches the time constant for recovery from inactivation of human layer 2/3 pyramidal neurons has been found to be $\sim 10 \text{ ms}$ ¹⁷). The red dotted line, in the inset of the figure shown below, corresponds to a mono-exponential curve with a time constant of 10 ms, to illustrate the suggested relationship between Na_v channel kinetics and axonal AP amplitude.

- Half-duration of axonal APs decreased with increasing duration of hyperpolarizations but was still detectable even after 1 s long hyperpolarizations. This is in line with the slow recovery from inactivation of axonal K_v1 channels (see response to point 4 and 5).

- Taken together, we conclude that due to the kinetics of the ion channels involved in the generation of axonal APs, a sequence of a longer depolarization followed by a brief hyperpolarization is the optimal stimulus to tune AP shape and boost presynaptic strength. We combined the figure panels shown above into Supplementary Fig. 4.

- In the introduction-statement the Reviewer writes: „ *Whilst these experiments are interesting, it is likely that the hyperpolarization increased the availability of Na⁺ channels and thereby boosted action potentials during the subsequent depolarization.* “. We want to take the opportunity and respond to this statement. In line with what the Reviewer states, our data suggests that hyperpolarizations can recover inactivated sodium channels to increase axonal AP amplitudes. However, it is important to clarify that this mechanism only becomes functionally relevant when there are sodium channels that can be recovered. For instance, we also tested the effect of isolated hyperpolarizations from resting membrane potential followed by APs (i.e. without a preceding depolarization: ‘Hyperpol’ condition; see Response to Reviewer 1, point 1). Resting membrane potentials of axons had a mean value of -74.4 ± 5 mV (mean \pm s.d., we did not correct for liquid junction potential, which we measured to be -15.2 ± 0.5 mV, mean \pm s.d.). We found that the axonal AP amplitudes were only increased by this ‘Hyperpol’ condition, if the resting membrane potential of an axon was more depolarized than -77 mV (5/10 axons). No effect was found in axons with more negative membrane voltages.

- This suggests that isolated hyperpolarizations only lead to a recovery of voltage gated sodium channels (Na_v) in more depolarized axons, whereas nearly all Na_v s are available for activation in axons with more negative membrane potentials. This indicates that presynaptic reliability is particularly effective in situations where a depolarization (leading to a broadening of axonal APs) is followed by a hyperpolarization (restoring the reduced AP amplitudes), while isolated events are less effective (see response to point 5).

4. Did the authors attempt to determine if a decrease in $KV1$ channel function is likely to account for the increased action potential duration induced by the depolarization, particularly as there are reports (e.g. Vivekananda et al., PNAS, 2017) that suggest that $KV1.1$ subunits may not be necessary for subthreshold modulation of spike width in rodents?

- We thank the Reviewer for raising this key point. We fully agree that establishing the mechanism of axonal AP broadening following subthreshold depolarization is important and would increase the significance of the manuscript. As pointed out by the Reviewer, inactivation of voltage-gated potassium channels has been identified as a mechanism underlying AP broadening across different mammalian species. To establish whether this is also a plausible mechanism in human cortical axons, we performed technically challenging voltage-clamp experiments in the ‘whole-bleb’ recording configuration. These experiments revealed an outward potassium current that activated at negative membrane voltages. The voltage at which half of the conductance was activated amounted to -7 mV (we did not correct membrane voltages for the liquid junction potential of -15.2 ± 0.5 mV, mean \pm s.d.). The potassium currents displayed inactivation. By using 5 s long conditioning pulses we obtained steady-state inactivation kinetics. The sigmoidal fit of the voltage-inactivation relationship indicated half-maximal inactivation at -43 mV.

- Puff-application of dendrotoxin-I (DTX-I) blocked ~55% of the current. The DTX-I sensitive fraction of the current corresponded to the inactivating component. This suggests that the observed inactivating outward current is mediated by potassium channels containing K_v1 alpha subunits.

- To obtain time constants that characterize the time course of inactivation and recovery from inactivation, we used a protocol consisting of 3 s long test pulses to +20 mV (holding potential of -80 mV) followed by brief test pulses to +20 mV at variable intervals (100, 300, 1000, 3000 ms). Inactivation was slow and best approximated by a sum of two exponential functions with a fast ($\tau_1 = 120$ ms) and a slow ($\tau_2 = 1.3$ s) time constant. The recovery from inactivation after such prolonged steps also followed a slow time course ($\tau_1 = 111$ ms; $\tau_2 = 1.0$ s).

- Taken together, we found that DTX-I sensitive potassium channels exist in the axons of human layer 2 & 3 pyramidal neurons. These channels are almost fully available for activation at resting membrane voltages of the axon (~ -75 mV) but can undergo considerable inactivation in the voltage range to the AP-threshold (~ -45 mV, i.e. the subthreshold range).

- Therefore, we argue that subthreshold depolarizations of sufficient duration can cause inactivation of a fraction of these K_v1 channels, which will then cause broadening of axonal APs.
- The DTX-I-block does not allow to distinguish between channels containing $K_v1.1$ versus $K_v1.2$ alpha subunits. The Allen Brain Institutes transcriptomic dataset of the human middle temporal gyrus suggests that both transcripts (*KCNA1* & 2) are expressed in layer 2 & 3 pyramidal neurons¹⁸. Studies that investigated axonal AP broadening by subthreshold depolarizations in neocortical pyramidal neurons in rodents and ferrets found that this effect is predominantly mediated by inactivation of $K_v1.2$ channels¹⁹ (as opposed to $K_v1.1$ in hippocampal pyramidal neurons^{20,21}). While our experiments provide evidence, that inactivation of axonal K_v1 potassium channels is a plausible mechanism for AP broadening in human cortical neurons, they don't allow to disentangle the exact contributions of $K_v1.1$ versus $K_v1.2$ alpha subunits. To address this, thorough pharmacological testing would be necessary. While this is an interesting avenue for future research, we argue that it is beyond the scope of this study.
- We combined the figure panels shown above into a new figure (see revised manuscript: Fig. 2), added a results paragraph and updated the methods section accordingly.

5. Did the authors investigate as to why hyperpolarization did not reverse the effects on K^+ channels induced by depolarization. Some insight into this would enhance the significance of their findings.

- We fully share the Reviewers' opinion that addressing this question would increase the clarity of the manuscript. In line with data from rodents^{21,22} and ferrets^{19,23}, we found that recovery from inactivation of axonal K_v1 channels is slow (see response to point 4). Therefore, brief hyperpolarizations (< 1 s) will not completely recover inactivated K_v1 channels and will not abolish the effect of depolarization induced AP broadening.

- This phenomenon is also reflected in the experiments that we performed in response to the Reviewers' point 3 and summarized in Supplementary Fig. 4. The red dotted line in the inset of the figure shown below corresponds to the bi-exponential function obtained in the voltage-clamp recordings ($\tau_1 = 111$ ms; $\tau_2 = 1.0$ s), and illustrates the suggested relationship between K_v channel kinetics and axonal AP duration.

- The following paragraph was added to our manuscript:

- “While the amplitude was completely restored, the half-duration of axonal APs remained broadened compared to ‘Control’ condition (Fig. 3c). This result is in line with the slow recovery from inactivation of axonal K_v1 potassium channels (Fig. 2d). In summary: When longer depolarizations are followed by shorter hyperpolarizations, fast recovery from inactivation of Na_vs (time constant of ~10 ms⁵⁹) outpaces slow recovery of axonal K_v1 channels, leading to tuned axonal APs (Supplementary Fig. 4).”

Remark to both Reviewers

A peer-researcher contacted us after reading the preprint version of the manuscript and suggested a discussion paragraph on extracellular Ca^{2+} concentration and presynaptic reliability of intracortical synapses during UP/DOWN-states *in-vivo*. We agree that adding such a paragraph is necessary to put our findings into the correct context. We modified the second discussion paragraph to address this:

- “*In-vivo* UP/DOWN states during SWA are complex network phenomena involving an intricate balance of excitation and inhibition^{20,23,24}. From the perspective of presynaptic terminals in the proximal axon, a central aspect of UP and DOWN states is that they give rise to membrane potential de- and hyperpolarizations, which can reach these terminals through passive propagation. Membrane potential de- and hyperpolarizations during UP- and DOWN-states passively spread along the axon and can reach proximal presynaptic terminals (Fig. 1f)⁴⁵. In this study, we showed that such presynaptic membrane potential changes modulate axonal AP-shape and hence synaptic strength-reliability between human layer 2 & 3 pyramidal neurons. While UP-state-like depolarizations enhance transmission presynaptic reliability, sequences of UP→DOWN states further increase synaptic transmission, with a presumed maximum after brief DOWN states (Fig. 4d-g). Besides AP-shape changes, other factors are known to affect presynaptic reliability during SWA. Notably, extracellular calcium concentration ($[\text{Ca}^{2+}]_o$) fluctuates during *in-vivo* SWA, de- and increasing during UP- and DOWN-states, respectively⁷⁰. The decrease of $[\text{Ca}^{2+}]_o$ during UP-states likely outweighs AP-broadening, as a net-decrease in synaptic reliability is observed with prolonged UP-states *in-vivo*⁷⁰. Conversely, the increased $[\text{Ca}^{2+}]_o$ during DOWN-states boosts presynaptic reliability, adding to the effect of tuned APs observed after UP→DOWN sequences (Fig. 4d-g). Thus, the ‘window-of-opportunity’ for increased presynaptic potency *in-vivo* is presumably confined to the DOWN-to-UP transition during UP→DOWN→UP cycles, as APs are ideally tuned (Fig 5i) and $[\text{Ca}^{2+}]_o$ is still high⁷¹. Since DOWN-to-UP transitions occur synchronously in numerous neurons during sleep^{19,72}, transmission should be enhanced at multiple synapses simultaneously. Therefore, these transition periods of SWA likely represent time windows of broadly increased presynaptic potency in supragranular pyramidal neuron circuits.”

References:

- 1 Neher, E. Correction for liquid junction potentials in patch clamp experiments. *Methods Enzymol* **207**, 123-131 (1992). [https://doi.org:10.1016/0076-6879\(92\)07008-c](https://doi.org:10.1016/0076-6879(92)07008-c)
- 2 Dickey, C. W. *et al.* Travelling spindles create necessary conditions for spike-timing-dependent plasticity in humans. *Nat Commun* **12**, 1027 (2021). <https://doi.org:10.1038/s41467-021-21298-x>
- 3 Vaz, A. P., Wittig, J. H., Jr., Inati, S. K. & Zaghoul, K. A. Replay of cortical spiking sequences during human memory retrieval. *Science* **367**, 1131-1134 (2020). <https://doi.org:10.1126/science.aba0672>
- 4 Andrillon, T. *et al.* Sleep spindles in humans: insights from intracranial EEG and unit recordings. *J Neurosci* **31**, 17821-17834 (2011). <https://doi.org:10.1523/JNEUROSCI.2604-11.2011>
- 5 Hahn, M. A., Heib, D., Schabus, M., Hoedlmoser, K. & Helfrich, R. F. Slow oscillation-spindle coupling predicts enhanced memory formation from childhood to adolescence. *Elife* **9** (2020). <https://doi.org:10.7554/eLife.53730>
- 6 Staresina, B. P. *et al.* Hierarchical nesting of slow oscillations, spindles and ripples in the human hippocampus during sleep. *Nat Neurosci* **18**, 1679-1686 (2015). <https://doi.org:10.1038/nn.4119>
- 7 Helfrich, R. F., Mander, B. A., Jagust, W. J., Knight, R. T. & Walker, M. P. Old Brains Come Uncoupled in Sleep: Slow Wave-Spindle Synchrony, Brain Atrophy, and Forgetting. *Neuron* **97**, 221-230 e224 (2018). <https://doi.org:10.1016/j.neuron.2017.11.020>
- 8 Helfrich, R. F. *et al.* Bidirectional prefrontal-hippocampal dynamics organize information transfer during sleep in humans. *Nat Commun* **10**, 3572 (2019). <https://doi.org:10.1038/s41467-019-11444-x>
- 9 Cohen, J. *Statistical Power Analysis For The Behavioral Sciences*. 2nd edn, (Routledge, New York, 1988).
- 10 Berg, J. *et al.* Human neocortical expansion involves glutamatergic neuron diversification. *Nature* **598**, 151-158 (2021). <https://doi.org:10.1038/s41586-021-03813-8>
- 11 Campagnola, L. *et al.* Local connectivity and synaptic dynamics in mouse and human neocortex. *Science* **375**, eabj5861 (2022). <https://doi.org:10.1126/science.abj5861>
- 12 Hunt, S. *et al.* Strong and reliable synaptic communication between pyramidal neurons in adult human cerebral cortex. *Cereb Cortex* **33**, 2857-2878 (2023). <https://doi.org:10.1093/cercor/bhac246>
- 13 Peng, Y. *et al.* Directed and acyclic synaptic connectivity in the human layer 2-3 cortical microcircuit. *Science* **384**, 338-343 (2024). <https://doi.org:10.1126/science.adg8828>
- 14 Steriade, M., Timofeev, I. & Grenier, F. Natural waking and sleep states: a view from inside neocortical neurons. *J Neurophysiol* **85**, 1969-1985 (2001). <https://doi.org:10.1152/jn.2001.85.5.1969>
- 15 Chauvette, S., Volgushev, M. & Timofeev, I. Origin of active states in local neocortical networks during slow sleep oscillation. *Cereb Cortex* **20**, 2660-2674 (2010). <https://doi.org:10.1093/cercor/bhq009>
- 16 Chauvette, S., Crochet, S., Volgushev, M. & Timofeev, I. Properties of slow oscillation during slow-wave sleep and anesthesia in cats. *J Neurosci* **31**, 14998-15008 (2011). <https://doi.org:10.1523/JNEUROSCI.2339-11.2011>
- 17 Wilbers, R. *et al.* Human voltage-gated Na(+) and K(+) channel properties underlie sustained fast AP signaling. *Sci Adv* **9**, eade3300 (2023). <https://doi.org:10.1126/sciadv.ade3300>
- 18 Siletti, K. *et al.* Transcriptomic diversity of cell types across the adult human brain. *Science* **382**, eadd7046 (2023). <https://doi.org:10.1126/science.add7046>
- 19 Shu, Y., Yu, Y., Yang, J. & McCormick, D. A. Selective control of cortical axonal spikes by a slowly inactivating K⁺ current. *Proc Natl Acad Sci U S A* **104**, 11453-11458 (2007). <https://doi.org:10.1073/pnas.0702041104>
- 20 Vivekananda, U. *et al.* Kv1.1 channelopathy abolishes presynaptic spike width modulation by subthreshold somatic depolarization. *Proc Natl Acad Sci U S A* **114**, 2395-2400 (2017). <https://doi.org:10.1073/pnas.1608763114>
- 21 Bialowas, A. *et al.* Analog modulation of spike-evoked transmission in CA3 circuits is determined by axonal Kv1.1 channels in a time-dependent manner. *Eur J Neurosci* **41**, 293-304 (2015). <https://doi.org:10.1111/ejn.12787>

- 22 Kole, M. H., Letzkus, J. J. & Stuart, G. J. Axon initial segment Kv1 channels control axonal action potential waveform and synaptic efficacy. *Neuron* **55**, 633-647 (2007).
<https://doi.org:10.1016/j.neuron.2007.07.031>
- 23 Shu, Y., Hasenstaub, A., Duque, A., Yu, Y. & McCormick, D. A. Modulation of intracortical synaptic potentials by presynaptic somatic membrane potential. *Nature* **441**, 761-765 (2006).
<https://doi.org:10.1038/nature04720>

REVIEWERS' COMMENTS

Reviewer #1 (Remarks to the Author):

The paper from Jörg Geiger's group is extremely impressive. This round of adequate revision has further increased its impact. I have no further comment. Congratulations.

Reviewer #2 (Remarks to the Author):

The manuscript shows for the first time that subthreshold signals mimicking up and down states modulate synaptic transmission in human neocortical neurons. The authors show that inactivation of KV1 currents is likely to, at least in part, mediate the action potential broadening that occurs following subthreshold depolarisations in human neurons. The manuscript is well-written and the methods are very clear. I have no further comments.